# On the Use of Drilling Degrees of Freedom to Stabilise the Augmented Finite Element Method

**Simon Essongue** [1,*] **, Guillaume Couégnat** [2] **and Eric Martin** [2]

[1] Arts et Métiers Institute of Technology, Université de Bordeaux, CNRS, INRA, Bordeaux INP, HESAM Université, I2M UMR 5295, 33170 Bordeaux, France

[2] Laboratoire des Composites Thermostructuraux, UMR5801 CNRS, University Bordeaux, Safran, CEA, 33600 Bordeaux, France

* Correspondence: simon.essongue-boussougou@u-bordeaux.fr

**Abstract:** The augmented finite element method (AFEM) embeds cracks within solid elements. These cracks are modelled without additional degrees of freedom thanks to a dedicated static condensation process. However, static condensation can induce a lack of constraint problem, resulting in singular stiffness matrices. To address this issue, we propose a new method called the stabilised augmented finite element method (SAFEM), which produces non-singular stiffness matrices. We conducted 2D experiments involving stationary traction-free cracks and propagating cohesive discontinuities to compare the performance of the SAFEM with the AFEM. The SAFEM outperforms the AFEM in modelling traction-free cracks.

**Keywords:** augmented finite element method; finite element analysis; drilling degrees of freedom; cohesive zone modelling; crack modelling; strong discontinuities; embedded finite element method





## 1. Introduction

In-plane rotational degrees of freedom (DOFs) are often referred to as "drilling degrees of freedom" in membrane finite elements. Their development started in the 1960s and served two purposes. Firstly, drilling DOFs improve the performance of linear membrane elements by allowing the manufacture of elements of intermediate computational cost and accuracy between linear and quadratic elements [1–4]. Secondly, when classical membrane elements and standard plate-bending elements are combined to form a shell element, each node possesses three displacement DOFs and two rotational DOFs in the local coordinate system of the shell. The presence of a third rotational DOF is attractive because it prevents the singularity of the stiffness matrix that occurs if shells meeting at a node are co-planar as outlined in Chap. 13.5 of Ref. [5].

In this paper, we explore a novel application of drilling DOFs: stabilising the augmented finite element method (AFEM). The AFEM was developed by Yang and coworkers as a numerical technique to embed weak or strong discontinuities within standard finite elements [6,7]. By embedding discontinuities inside elements, the AFEM simplifies their modelling because they no longer need to conform to the elements boundaries as required in the finite element method (FEM). The AFEM has been widely used in structural analysis to model cracks, with successful applications, including unstable crack propagation [8], crack growth under thermomechanical loading [9], dynamic crack propagation [10], three-dimensional studies of composite materials [11] or the large deformation of cracked shells [12]. The method was implemented as a user element in Abaqus and proved to be 50 times faster than the phantom-node method (PNM) natively implemented in this software [13]. The AFEM offers a significant advantage over other partition of unity methods, such as the PNM or the extended finite element method [14,15] (XFEM): it does not require adding global unknowns to model discontinuities. This allows the AFEM to represent an arbitrary number of growing cracks while keeping the size of the global stiffness matrix

constant. However, the AFEM does have a major drawback: it requires handling stiffness matrices that become singular in some situations [6,16]. The AFEM originators briefly discussed this issue in Ref. [6], but no satisfactory solution was provided. We thus previously proposed a dedicated procedure to handle these singular matrices [16], which is recalled in Section 3.4. As far as we know, our proposed strategy is the only solution for using augmented elements with traction-free cracks published to date.

The use of drilling DOFs proposed in this work improves the AFEM by preventing the occurrence of singular stiffness matrices, eliminating the need for a dedicated procedure to handle them. Additionally, drilling DOFs substantially increases the accuracy of the AFEM. The improved AFEM with drilling DOFs will be referred to as the stabilised augmented finite element method (SAFEM) in this work. Numerical experiments conducted with an in-house code [16,17] demonstrate that the SAFEM outperforms the AFEM when cracks are traction-free. The results obtained with cohesive cracks are of similar accuracy.

An outline of this paper is as follows. Section 2 provides a brief review of the formulation of finite elements with drilling DOFs and includes some accuracy checks. Section 3 presents a summary of the general AFEM framework for modelling cohesive cracks and demonstrates how drilling DOFs can be introduced within augmented elements. In Section 4, we perform multiple experiments to evaluate the accuracy of the SAFEM and compare it to the original AFEM.

## 2. Finite Elements with Drilling Degrees of Freedom

According to MacNeal and Harder, membrane finite elements with drilling degrees of freedom (DOFs) were proposed several times between 1960 and 1980, but these attempts were mostly unsuccessful [3]. The distinct approaches taken by Bergan and Felippa [18] and Allman [19], as well as the mathematical framework proposed by Hughes and Brezzi [20] paved the way for the successful formulations used today (see, for example, [21–23]). In this work, we will consider the approach proposed by Allman in 1984 to design membrane elements with in-plane rotational DOFs [19]. We chose Allman's elements because of their simplicity, but other formulations could have been considered as well. To keep the exposition to a reasonable length, only 2D membrane elements will be investigated, but Allman's approach is equally applicable to 3D elements, as exemplified in Refs. [24,25]. We begin by recalling the formulation of Allman's elements and check their accuracy. A word of caution is in order: Allman derived other elements with drilling DOFs in 1988 [2], which are different from those considered here; indeed, they are incompatible. Hence, one must bear in mind that this work only employs Allman's 1984 compatible elements.

The goal of the present section is to benchmark Allman's elements against "classical" finite elements. It is a fundamental step. Indeed, the SAFEM uses Allman's elements and will inherit their properties, such as the convergence rate and sensitivity to distortions. To the best of our knowledge, the results we are about to present regarding Allman's 1984 elements were yet to be found in the literature.

### 2.1. Design and Formulation of Allman's Elements

Allman derived a triangular element with drilling DOFs through intuitive geometrical concepts [19]. Cook and coworkers re-interpreted Allman's work as the result of applying a coordinate transformation to elements with midside nodes and applied it to quadrilateral, tetrahedral and hexahedral elements [1,24,25]. The approach can be summarised as follows: one starts with a higher-order isoparametric element with straight edges and midside nodes (e.g., the six-node linear strain triangle or the eight-node Serendipity quadrangle). Then, one "transforms" the midside translational DOF into drilling DOFs located at the element's corners through master–slave constraints. Felippa called the resulting operation a "retrofitting" procedure in Ref. [21].

This section focuses on the retrofitting fabrications of the following: (i) a three-node triangle with drilling DOFs, coined T3A (where "A" stands for Allman), from a six-node

linear strain triangle (T6) and (ii) a four-node quadrilateral with drilling DOFs (Q4A) from an eight-node biquadratic quadrilateral (Q8), see Figure 1.

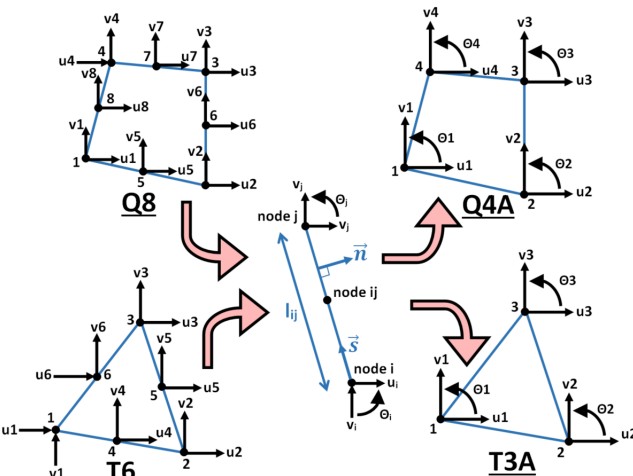

**Figure 1.** Fabrication of Allman's elements. Step 1: choice of an element with straight edges and midside nodes (e.g., the T6 or the Q8), step 2: transformation of the DOFs supported by the midside nodes into drilling DOFs, and step 3: suppression of the midside nodes to obtain Allman's elements (e.g., the T3A or the Q4A).

Allman's elements are obtained by choosing specific polynomial orders for the components of the displacement vector along each edge of finite elements with midside nodes: the normal and tangential components of the displacement field along the edges, $u_n$ and $u_t$, are quadratic and linear, respectively. Following Allman's approach, the expressions for $u_n$ and $u_t$ are [19]:

$$u_n(s) = (1 - \frac{s}{l_{ij}})u_{n_i} + \frac{s}{l_{ij}}u_{n_j} + \frac{4s}{l_{ij}}(1 - \frac{s}{l_{ij}})u_{n_{ij}} \tag{1}$$

$$u_t(s) = (1 - \frac{s}{l_{ij}})u_{t_i} + \frac{s}{l_{ij}}u_{t_j} \tag{2}$$

where $l_{ij}$ is the length of the edge between nodes $i$ and $j$, $s$ is the curvilinear abscissa along the edges of the elements and $i$ and $j$ are indexes associated with the element nodes (cf. Figure 1). For instance, $u_{n_j}$ is the degree of freedom associated with the normal component of the displacement field at node $j$. The use of both indexes $i$ and $j$ means that one considers the degree of freedom associated with a midside node, e.g., $u_{t_{ij}}$, is the DOF associated with the tangential component of the displacement field at the midside node located between nodes $i$ and $j$. The degrees of freedom supported by the midside nodes (i.e., $u_{n_{ij}}$ in Equation (1)) are then replaced by rotational DOFs supported by the apex nodes. Vertex rotations are introduced for that purpose and the following relation is made use of:

$$u_{n_{ij}} = \frac{l_{ij}}{8}(\Theta_j - \Theta_i) \tag{3}$$

where $\Theta_i$ are the so-called drilling DOFs representing the vertex rotation at node $i$. Despite the use of drilling DOFs, Allman's formulation does not seek to interpolate a rotation field. Instead, it only employs vertex rotations to enhance the displacement field. This is in contrast to elements with drilling DOFs, which interpolate the rotation field and displacement field independently (as seen in [20,23]). Furthermore, it is important to note that the vertex rotations introduced in Allman's approach are not true rotations as defined in continuum mechanics, but they are closely related. Indeed, the following relation holds for the T3A [19]:

$$\Omega_i - \frac{1}{3}(\Omega_1 + \Omega_2 + \Omega_3) = \frac{3}{4}\left(\Theta_i - \frac{1}{3}(\Theta_1 + \Theta_2 + \Theta_3)\right) \text{ for } i \in [1,2,3] \tag{4}$$

$$\text{with } \Omega(x,y) = \frac{1}{2}\left(\frac{\partial v(x,y)}{\partial x} - \frac{\partial u(x,y)}{\partial y}\right) \tag{5}$$

where $\Omega$ is the true rotation field; $\Omega_i$ is the value of the true rotation field at node $i$; and $u$ and $v$ are the horizontal and vertical components of the displacement field, respectively. Equation (4) reveals that the differences between the true rotations at the vertices ($\Omega_i$) and the average true rotation are directly proportional to the discrepancy between the vertex rotations ($\Theta_i$) and their mean. Equations (1)–(3) highlight how the translational DOF supported by the midside nodes of higher-order elements can be turned into vertex rotation DOFs. Consequently, the midside nodes and their associated DOF are no longer required (cf. Figure 1) and the following relations hold:

$$\{u_1, v_1, ..., u_6, v_6\}^t = [T_{\text{T6-T3A}}]\{u_1, v_1, \Theta_1, ..., u_3, v_3, \Theta_3\}^t \tag{6}$$

$$\{u_1, v_1, ..., u_8, v_8\}^t = [T_{\text{Q8-Q4A}}]\{u_1, v_1, \Theta_1, ..., u_4, v_4, \Theta_4\}^t \tag{7}$$

where $[T_{\text{T6-T3A}}]$ and $[T_{\text{Q8-Q4A}}]$ are the so-called transformation matrices that solely depend on the coordinates of the element vertices [1,3]. Closed-form expressions for these matrices are provided in Appendix A. Once the transformation matrices are known, the stiffness matrices of Allman's elements can be readily computed from the stiffness matrices of the higher-order elements they are based on. Specifically, if $[K_{\text{T6}}]$, $[K_{\text{Q8}}]$, $[K_{\text{T3A}}]$ and $[K_{\text{Q4A}}]$ are the stiffness matrices of the T6, Q8, T3A and Q4A, respectively, then:

$$[K_{\text{T3A}}] = [T_{\text{T6-T3A}}]^t[K_{\text{T6}}][T_{\text{T6-T3A}}] \tag{8}$$

$$[K_{\text{Q4A}}] = [T_{\text{Q8-Q4A}}]^t[K_{\text{Q8}}][T_{\text{Q8-Q4A}}] \tag{9}$$

The elements resulting from the use of Equations (8) and (9) are compatible, satisfy the patch tests, employ second-order shape functions and only involve DOFs located at corner nodes. Allman's elements can be integrated using the same quadrature rules as their underlying higher-order elements. In this study, we integrate the stiffness matrices of the T3A and the T6 with three integration points, while the Q4A and the Q8 make use of a $3 \times 3$ Gaussian integration scheme. These quadratures fully integrate the corresponding higher-order elements, but the associated Allman's elements are rank deficient. There exists a spurious deformation mode with identical vertex rotations (i.e., $\Theta_1 = \Theta_2 = ... = \Theta_n$) whatever the chosen integration scheme [3]. Several strategies have been proposed to deal with it: (i) constraining a single drilling DOF of the entire mesh to prevent the appearance of the spurious mode or (ii) adding a penalty matrix to the stiffness matrices in order to restore their full rank [3,4,24,25]. These two options will be investigated. The penalty stiffness used to prevent rank deficiencies of Allman's elements was designed by Macneal and coworkers and is given by (see Equations (46)–(49) of Ref. [3]):

$$[K_{ij}^{penalty}] = \gamma V G \frac{\partial^2(\Theta_\Delta^2)}{\partial q_i \partial q_j} \tag{10}$$

$$\text{with } \Theta_\Delta = \frac{1}{n}\sum_{i=1}^{n}(\Theta_i - \Omega_0) \text{ and } \Omega_0 = \frac{1}{2}\left(\frac{\partial v(x,y)}{\partial x} - \frac{\partial u(x,y)}{\partial y}\right)_0 \tag{11}$$

where $\gamma$ is the dimensionless penalty parameter, $V$ is the element volume, $G$ is the shear modulus, n is the number of nodes of Allman's elements, $\Theta_i$ is the drilling DOF at node $i$, $q_i$ is the $i$th component of the DOF vector (e.g., $q_1 = u_1$, $q_2 = v_1$ or $q_3 = \Theta_1$, cf. Equation (6)) and $\Omega_0$ is the true rotation computed at the centroid of the element thanks to the shape function derivatives. Unlike the stiffness matrices, the penalty matrix is computed without using quadrature rules and instead employs a closed-form expression. Equations (10) and (11) show that the penalty matrix elastically ties the weighted average

of the corner rotations to the in-plane rotation at the centre of the element, with no effect on the translational DOF. Choosing an appropriate value for the dimensionless coefficient $\gamma$ is crucial, as it involves a compromise between competing effects. If $\gamma$ is too large, any accuracy improvement resulting from the inclusion of rotations will be lost. On the other hand, if it is too low, the resulting matrices will become rank deficient. In Ref. [3], Macneal and coworkers suggested a value of $10^{-6}$ for $\gamma$.

### 2.2. Accuracy of Allman's Elements

Allman's elements have been employed in various studies, as documented in the literature [1–4,19]. However, the influence of the dimensionless coefficient $\gamma$, which controls the level of stabilisation provided to these elements, has received little attention. Moreover, to the best of our knowledge, the convergence rate of Allman's elements in the energy norm has never been examined. To gain more insight into the accuracy of these elements, we conduct two series of numerical experiments. Material and geometric parameters will be given without units, as is customary in computational mechanics [18,21,26].

#### 2.2.1. Short Cantilever Beam Test

The short cantilever beam test is employed to study the coarse mesh accuracy of Allman's elements, their sensitivity to distortions and the effect of the dimensionless coefficient $\gamma$. The short cantilever beam test consists of applying a parabolic distribution of shear stresses at the tip of a beam while the left-hand side is fully clamped (Figure 2). Following Felippa [21], the problem is considered in a non-dimensional form, and we use Young's modulus $E = 2660$ and a Poisson's ratio of $\nu = 0.2$ and apply a vertical force $P = 100$, so that the exact vertical displacement at the centre of the end-loaded cross section is $v_{analytical} = \frac{L_{beam}^3 P}{3EI_z} + \frac{(4+5\nu)(L_{beam}P)}{2EH_{beam}} = 10$, where $L_{beam}$ and $H_{beam}$ are the beam's dimensions, provided in Figure 2, and $I_z = \frac{L_{beam}H_{beam}^3}{12}$. A thickness of 1 and a plane stress state are considered.

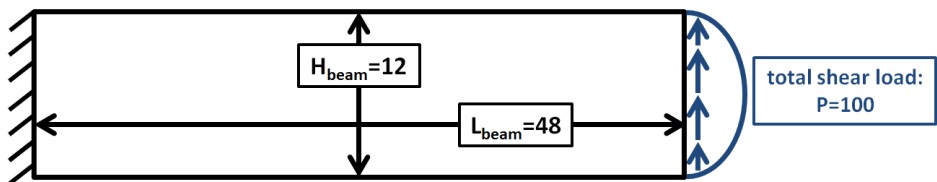

**Figure 2.** Cantilever beam under end shear loading.

The numerical experiment is run with Allman's elements and the underlying higher-order finite elements depicted in Figure 1. For comparison purposes, we also benchmark the $2 \times 2$ Gauss-integrated bilinear quadrilateral (Q4) and the constant strain triangle (T3). Regular and distorted meshes with one element through the thickness are employed (Figure 3).

The resulting vertical displacements at the centre of the end-loaded cross section are compared with the analytical solution value in Table 1.

The following conclusions can be drawn from the numerical results: (i) the accuracy of elements with drilling DOFs lies in between those of the underlying higher-order elements and the linear elements, (ii) the use of drilling DOFs substantially increases the performance of classical linear elements, (iii) Allman's elements are less sensitive to distortions than linear elements and (iv) the penalty stiffness does not affect the results when the recommended value $\gamma = 10^{-6}$ is used.

These results are highly significant and indicate a positive outlook for the incorporation of Allman's elements in the AFEM framework. As stressed in Section 3, the stabilised augmented finite element method (SAFEM) will rely on Allman's elements in arbitrarily distorted configurations. Therefore, had the elements been highly sensitive to distortions, they would have been unsuitable for the method.

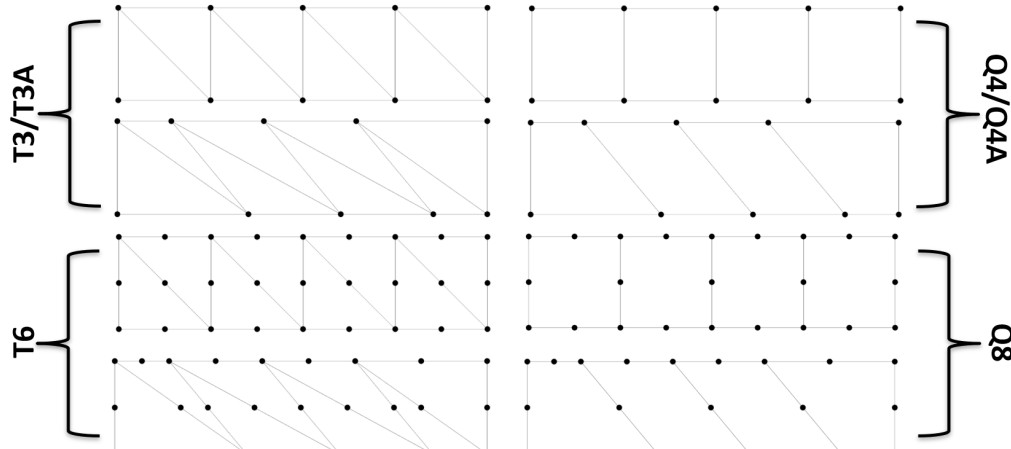

**Figure 3.** Regular and distorted discretisations used to run the cantilever beam test.

**Table 1.** Comparison of the computed vertical displacements at the tip of the short cantilever beam.

|  | Regular Mesh | Distorted Mesh | Number of DOFs |
|---|---|---|---|
| exact solution | 10. | 10. | - |
| T3 | 2.55017 | 1.4409 | 20 |
| T3A $\gamma = 0$. T3A $\gamma = 10.^{-6}$ | 7.65647 | 5.60141 | 30 |
| T6 | 9.85754 | 9.35355 | 54 |
| Q4 | 6.85744 | 4.02667 | 20 |
| Q4A $\gamma = 0$. | 9.45564 | 9.15315 | 30 |
| Q4A $\gamma = 10.^{-6}$ | 9.45511 | 9.15279 | 30 |
| Q8 | 9.88888 | 9.91559 | 46 |

### 2.2.2. Convergence Rate in the Energy Norm

To the best of the authors' knowledge, no previous studies have investigated the rate of convergence of Allman's elements. As a result, the speed at which error measures decrease with mesh refinement when using these elements remains unknown. In this section, we focus on the error in the energy norm, as this measure is crucial to several fundamental properties of the FEM, such as Galerkin orthogonality (see Chap. 4.3 of Ref. [27]). The objective of this section is to investigate the convergence behaviour of Allman's elements in the energy norm through numerical experiments. We will also compare the performances of both triangular and quadrilateral elements with the finite element method. Before presenting the numerical experiment, we introduce the error in the energy norm and some related quantities. Let **u** be the exact solution of a mechanical problem and **û** an approximate solution, and the error **e** is:

$$\mathbf{e} = \mathbf{u} - \hat{\mathbf{u}} \tag{12}$$

Let $\Omega$ be the volume occupied by the structure of interest, and the relative error in the energy norm is:

$$||\mathbf{e}|| = \left( \frac{\int_\Omega (\epsilon - \hat{\epsilon}) : C : (\epsilon - \hat{\epsilon}) \, d\Omega}{\int_\Omega \epsilon : C : \epsilon \, d\Omega} \right)^{\frac{1}{2}} \tag{13}$$

where $\epsilon$ is the exact strain field, $\hat{\epsilon}$ is the finite element strain field and $C$ is the elasticity stiffness tensor. Galerkin orthogonality allows for simplifying the computation of the relative error in the energy norm:

$$||\mathbf{e}|| = \left( \frac{\int_\Omega \epsilon : C : \epsilon \, d\Omega - \int_\Omega \hat{\epsilon} : C : \hat{\epsilon} \, d\Omega}{\int_\Omega \epsilon : C : \epsilon \, d\Omega} \right)^{\frac{1}{2}} \tag{14}$$

To investigate the convergence rate of Allman's elements, we perform a test involving a circular beam subjected to end shear in a state of plane stress. The geometry and loading of the problem are shown in Figure 4. The displacements on the boundaries are prescribed:

$$u_\theta(r, \theta = 90°) = u_r(r = a, \theta = 90°) = 0 \text{ and } u_r(r, \theta = 0°) = u_0 \tag{15}$$

Following Zienkiewicz (example 2.4 of Ref. [5]), the following geometric and material properties are considered: $a = 5$, $b = 10$, $E = 10,000$ and $\nu = 0.25$. A displacement of magnitude $u_0 = -0.01$ is imposed on the right end of the circular beam. The above values lead to the following closed-form strain energy [5]:

$$\int_\Omega \epsilon : C : \epsilon \, d\Omega = \frac{1}{\pi}(log(2) - 0.6) \tag{16}$$

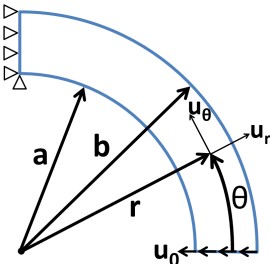

**Figure 4.** End-loaded circular beam: problem geometry and boundary conditions.

The finite element solution to the problem is obtained using uniform meshes made of Allman's elements and the underlying higher-order finite elements: the Q8 and the T6. For comparison, we also benchmarked the $2 \times 2$ Gauss-integrated bilinear quadrilateral (Q4) as well as the constant strain triangle (T3) (cf. Figure 5). Elements with drilling DOFs are sometimes referred to as "higher-order" ones [21,28] because they derive from second-order elements. However, as shown in Figure 5, Allman's elements converge with an error of $\mathcal{O}(h)$ in the energy norm, just like linear finite elements. This convergence rate could be expected because the tangential displacement along the edges of Allman's elements is only linear (cf. Equation (2)). Thus, while Allman's elements are more accurate than standard finite elements that share the same topology, they cannot be considered as "higher-order" elements. One can also notice that the penalty stiffness does not affect the results when the recommended value $\gamma = 10^{-6}$ is employed. This value will be used consistently throughout this work.

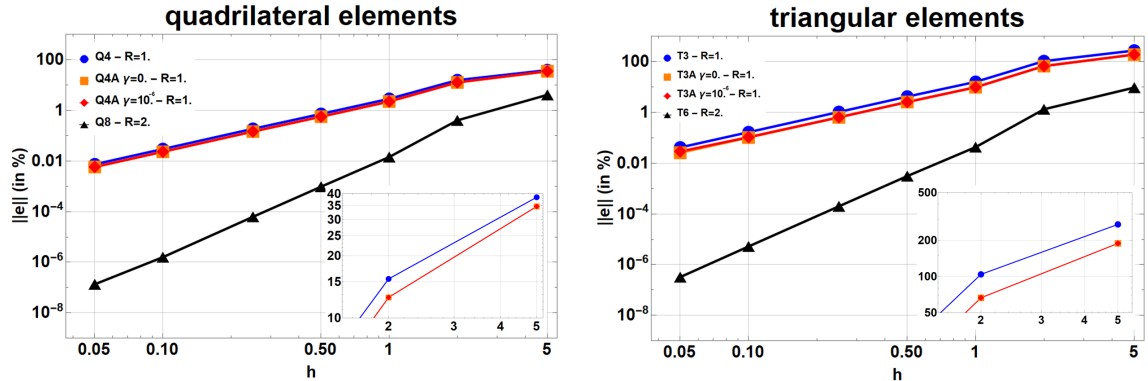

**Figure 5.** Relative error in the energy norm, $||\mathbf{e}||$, as a function of the mesh size, $h$, and associated convergence rate, $R$, obtained with linear, quadratic and Allman's finite elements.

### 3. Augmented Finite Elements with Drilling Degrees of Freedom

This self-contained section presents the derivation of the augmented finite element method (AFEM), the need for stabilising the method and the incorporation of drilling DOFs for achieving stabilisation. In Section 4, the resulting stabilised augmented finite element method (SAFEM) will be benchmarked against the AFEM. In this study, we only consider strong discontinuities because stabilisation is not required for modelling weak discontinuities [6,16,29].

#### 3.1. Strong Form

The reference situation to be considered is schematised in Figure 6. Let $\Omega$ be the domain occupied by a solid. A material point inside the domain is labelled as $\mathbf{x} \in \Omega$. A cohesive strong discontinuity surface $\Gamma_c = \Gamma_c^+ \cup \Gamma_c^-$ splits $\Omega$ into two subdomains, $\Omega^+$ and $\Omega^-$. The prescribed external tractions $\mathbf{t}_{\mathbf{ext}}^+$ and $\mathbf{t}_{\mathbf{ext}}^-$ are applied on boundary $\Gamma_t = \Gamma_t^+ \cup \Gamma_t^-$, whereas the displacements $\bar{\mathbf{u}}^+$ and $\bar{\mathbf{u}}^-$ are imposed on boundary $\Gamma_u = \Gamma_u^+ \cup \Gamma_u^-$. The domains on both sides of the discontinuity are assumed to be elastic and homogeneous, yet $\Omega^+$ and $\Omega^-$ can be made of different materials. We further assume small strain and displacement conditions.

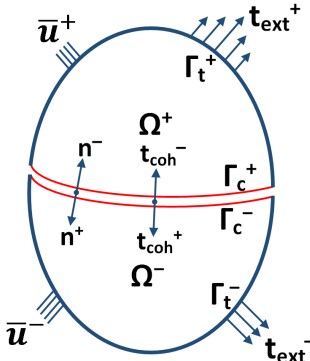

**Figure 6.** Solid body crossed by a cohesive strong discontinuity surface.

In the absence of body forces, the field equations governing the boundary value problem obey the following relations:

$$
\begin{array}{llll}
\nabla\sigma^+(\mathbf{x}) = \mathbf{0} & \mathbf{x} \in \Omega^+ & \nabla\sigma^-(\mathbf{x}) = \mathbf{0} & \mathbf{x} \in \Omega^- \quad (17)\\[4pt]
\sigma^+(\mathbf{x})\cdot\mathbf{n}^+(\mathbf{x}) = \mathbf{t}_{\mathbf{ext}}^+(\mathbf{x}) & \mathbf{x} \in \Gamma_t^+ & \sigma^-(\mathbf{x})\cdot\mathbf{n}^-(\mathbf{x}) = \mathbf{t}_{\mathbf{ext}}^-(\mathbf{x}) & \mathbf{x} \in \Gamma_t^- \quad (18)\\[4pt]
\mathbf{u}^+(\mathbf{x}) = \bar{\mathbf{u}}^+(\mathbf{x}) & \mathbf{x} \in \Gamma_u^+ & \mathbf{u}^-(\mathbf{x}) = \bar{\mathbf{u}}^-(\mathbf{x}) & \mathbf{x} \in \Gamma_u^- \quad (19)\\[4pt]
\mathbf{t}_{\mathbf{coh}}^+(\mathbf{x}) = \sigma^+(\mathbf{x})\cdot\mathbf{n}^+(\mathbf{x}) & \mathbf{x} \in \Gamma_c^+ & \mathbf{t}_{\mathbf{coh}}^-(\mathbf{x}) = \sigma^-(\mathbf{x})\cdot\mathbf{n}^-(\mathbf{x}) & \mathbf{x} \in \Gamma_c^- \quad (20)
\end{array}
$$

$\sigma^+$ (resp. $\sigma^-$) is the stress field in $\Omega^+$ (resp. $\Omega^-$), $\mathbf{t}_{\mathbf{coh}}^+$ and $\mathbf{t}_{\mathbf{coh}}^-$ are the tractions along the discontinuity surface $\Gamma_c = \Gamma_c^+ \cup \Gamma_c^-$ and $\mathbf{n}^+$ (resp. $\mathbf{n}^-$) is the outward-pointing normal of $\Omega^+$ (resp. $\Omega^-$). Tractions $\mathbf{t}_{\mathbf{coh}}^+$ and $\mathbf{t}_{\mathbf{coh}}^-$ are related to the cohesive strong discontinuity opening, denoted as $\Delta\mathbf{u}$ and given by:

$$\Delta\mathbf{u}(\mathbf{x}) = \mathbf{u}^+(\mathbf{x}) - \mathbf{u}^-(\mathbf{x}) \quad \mathbf{x} \in \Gamma_c \tag{21}$$

Equilibrium imposes the following equalities:

$$\mathbf{t}_{\mathbf{coh}}^-(\mathbf{x}) = -\mathbf{t}_{\mathbf{coh}}^+(\mathbf{x}) = \mathbf{t}_{\mathbf{coh}}(\mathbf{x}) \quad \mathbf{x} \in \Gamma_c \tag{22}$$

The constitutive law and the strain–displacement equations for the two subdomains are:

$$\sigma^+(\mathbf{x}) = \mathbf{C}^+ : \epsilon^+(\mathbf{x}) \quad \mathbf{x} \in \Omega^+ \quad \epsilon^+(\mathbf{x}) = \frac{1}{2}(\nabla^T \mathbf{u}^+(\mathbf{x}) + \nabla \mathbf{u}^+(\mathbf{x})) \quad \mathbf{x} \in \Omega^+ \quad (23)$$

$$\sigma^-(\mathbf{x}) = \mathbf{C}^- : \epsilon^-(\mathbf{x}) \quad \mathbf{x} \in \Omega^- \quad \epsilon^-(\mathbf{x}) = \frac{1}{2}(\nabla^T \mathbf{u}^-(\mathbf{x}) + \nabla \mathbf{u}^-(\mathbf{x})) \quad \mathbf{x} \in \Omega^- \quad (24)$$

where $\mathbf{C}^+$ and $\mathbf{C}^-$ are the stiffness tensors of the subdomains $\Omega^+$ and $\Omega^-$, respectively. In this work, a weakly coupled and bilinear elasto-damage law governs the relationship between cohesive tractions $\mathbf{t_{coh}}$ and openings $\mathbf{\Delta u}$. It is important to recognise that the SAFEM and the AFEM can be employed with any shape of cohesive law (e.g., exponential or multi-linear). The traction separation law employed in this article is given by

$$\mathbf{t_{coh}} = [\mathbf{D_{coh/sec}}(d_n, d_t)]\mathbf{\Delta u} \quad (25)$$

where we introduced the scalar damage variables $d_n$ and $d_t$ associated with normal and tangential directions to the discontinuity surface. The matrix $[\mathbf{D_{coh/sec}}]$ is the secant stiffness matrix associated with the employed cohesive law. Because any cohesive law can be embedded within an augmented element, the detailed description of the employed traction–opening relationship is deferred to Appendix B. Recent work has demonstrated that the AFEM also allows interphases to be embedded within solid elements rather than interfaces [30].

Equations (17)–(25) define the problem strong form. The AFEM differs significantly from other strong discontinuity approaches, such as the numerous embedded finite element methods (EFEM) or the extended finite element method (XFEM) (see, e.g., Refs. [15,30–37]). Indeed, the cohesive internal tractions are treated similarly to the external tractions, and no discontinuous displacement or strain field is introduced. The above strong form actually shares many similarities with that of cohesive zone models, see, e.g., [38].

*3.2. Discretised Weak Form*

The spaces of the admissible continuous displacement fields $\mathcal{U}^+$ and $\mathcal{U}^-$ are defined by

$$\mathcal{U}^+ = \{\mathbf{u}^+ \in H1 : \mathbf{u}^+ = \bar{\mathbf{u}}^+ \text{ on } \Gamma_u^+\} \quad \mathcal{U}^- = \{\mathbf{u}^- \in H1 : \mathbf{u}^- = \bar{\mathbf{u}}^- \text{ on } \Gamma_u^-\} \quad (26)$$

where $H1$ is the space of functions with square-integrable derivatives, i.e., the Sobolev space of degree 1 defined by:

$$H1 = \{\mathbf{w} : \mathbf{w} \in L2 \,, D\mathbf{w} \in L2\} \quad \text{with} \quad L2 = \{\mathbf{w} : \int \mathbf{w}^2 dx < \infty\} \quad (27)$$

where $\mathbf{w}$ is a function and $D\mathbf{w}$ denotes its partial derivative.

Let us introduce the spaces of continuous test functions $\mathcal{V}^+$ and $\mathcal{V}^-$ defined by

$$\mathcal{V}^+ = \{\mathbf{v}^+ \in H1 : \mathbf{v}^+ = \mathbf{0} \text{ on } \Gamma_u^+\} \quad \mathcal{V}^- = \{\mathbf{v}^- \in H1 : \mathbf{v}^- = \mathbf{0} \text{ on } \Gamma_u^-\} \quad (28)$$

The weak forms of the equilibrium equations in the two subdomains $\Omega^+$ and $\Omega^-$ are stated as follows: find $\mathbf{u}^+ \in \mathcal{U}^+$ and $\mathbf{u}^- \in \mathcal{U}^-$ such that

$$\int_{\Omega^+} \sigma^+(\mathbf{u}^+(\mathbf{x})) : \epsilon^+(\mathbf{v}^+(\mathbf{x})) d\Omega = \int_{\Gamma_t^+} \mathbf{t_{ext}^+}(\mathbf{x}) \cdot \mathbf{v}^+(\mathbf{x}) d\Gamma - \int_{\Gamma_c^+} \mathbf{t_{coh}}(\mathbf{x}) \cdot \mathbf{v}^+(\mathbf{x}) d\Gamma \quad \forall \mathbf{v} \in \mathcal{V}^+ \quad (29)$$

$$\int_{\Omega^-} \sigma^-(\mathbf{u}^-(\mathbf{x})) : \epsilon^-(\mathbf{v}^-(\mathbf{x})) d\Omega = \int_{\Gamma_t^-} \mathbf{t_{ext}^-}(\mathbf{x}) \cdot \mathbf{v}^-(\mathbf{x}) d\Gamma + \int_{\Gamma_c^-} \mathbf{t_{coh}}(\mathbf{x}) \cdot \mathbf{v}^-(\mathbf{x}) d\Gamma \quad \forall \mathbf{v} \in \mathcal{V}^- \quad (30)$$

The above equations are referred to as the principle of virtual work (see, e.g., p. 78 of Ref. [39]). The left-hand sides of Equations (29) and (30) are the internal virtual work, and the right-hand sides are the virtual work carried out by the external forces and the tractions along the strong discontinuity surface. The application of the (Bubnov–)Galerkin method to Equations (29) and (30) leads to the following matrix equations (see, e.g., Chap. 2.8 [39]):

$$[L^+]\{d^+\} = \left\{ \begin{matrix} fext^+ \\ -fcoh^+ \end{matrix} \right\} \qquad [L^-]\{d^-\} = \left\{ \begin{matrix} fext^- \\ -fcoh^- \end{matrix} \right\} \tag{31}$$

where vector $\{d^+\}$ (resp. $\{d^-\}$) contains the DOF in $\Omega^+$ (resp. $\Omega^-$). $[L^+]$ and $[L^-]$ are the finite element stiffness matrices of subdomains $\Omega^+$ and $\Omega^-$, respectively. If, for example, $\Omega^+$ is a triangular subdomain, then $[L^+]$ would correspond to the stiffness matrix of a T3 element in the AFEM formulation or the stiffness matrix of a T3A element (as defined by Equation (8)) in the SAFEM formulation. The external forces, defined later in this section, appear on the right-hand side of Equation (31). The displacement field in $\Omega^+$ (resp. $\Omega^-$) is expressed thanks to the shape function matrix $[N^+]$ (resp. $[N^-]$):

$$\mathbf{u}^+(\mathbf{x}) = [N^+(\mathbf{x})]\{d^+\} \ \mathbf{x} \in \Omega^+ \qquad \mathbf{u}^-(\mathbf{x}) = [N^-(\mathbf{x})]\{d^-\} \ \mathbf{x} \in \Omega^- \tag{32}$$

The interpolations presented in Equation (32) are only valid for an element crossed by a strong discontinuity, i.e., an augmented element. Crack-free elements are standard finite elements that use a single shape function matrix $[N]$.

To better illustrate the formulation, we consider a triangular domain $\Omega$ divided into a quadrilateral subdomain ($\Omega^-$) and a triangular subdomain ($\Omega^+$), as depicted in Figure 7. With this configuration, the shape function matrix $[N^+]$ (resp. $[N^-]$) corresponds to a T3A (resp. Q4A) element with the SAFEM and a T3 (resp. Q4) element with the AFEM. The DOF vectors in Equation (32) are partitioned between those associated with the strong discontinuity surface, denoted as $\{dcoh^+\}$ and $\{dcoh^-\}$, and those associated with the surface of external tractions, $\{dext^+\}$ and $\{dext^-\}$, as follows:

$$\{d^+\} = \left\{ \begin{matrix} dext^+ \\ dcoh^+ \end{matrix} \right\} \qquad \{d^-\} = \left\{ \begin{matrix} dext^- \\ dcoh^- \end{matrix} \right\} \tag{33}$$

The DOF vector $\{dcoh^+\}$ (resp. $\{dcoh^-\}$) allows for expressing the strong discontinuity surface opening thanks to the shape functions $N^+$ (resp. $N^-$). The opening–displacement matrix of the cohesive surface $[B^{coh}]$ writes [38]

$$\Delta\mathbf{u}(\mathbf{x}) = \begin{bmatrix} -N^-(\mathbf{x}) & N^+(\mathbf{x}) \end{bmatrix} \left\{ \begin{matrix} dcoh^- \\ dcoh^+ \end{matrix} \right\} = [B^{coh}(\mathbf{x})] \left\{ \begin{matrix} dcoh^- \\ dcoh^+ \end{matrix} \right\} \qquad \mathbf{x} \in \Gamma_c \tag{34}$$

Matrices $[L^+]$ and $[L^-]$ in Equation (31) are the stiffness matrices of the two linear elastic subdomains and are given by

$$[L^+] = \begin{bmatrix} L_{11}^+ & L_{12}^+ \\ L_{21}^+ & L_{22}^+ \end{bmatrix} = \int_{\Omega^+} [B^+(\mathbf{x})]^t [\mathbf{C}^+][B^+(\mathbf{x})]d\Omega \tag{35}$$

$$[L^-] = \begin{bmatrix} L_{11}^- & L_{12}^- \\ L_{21}^- & L_{22}^- \end{bmatrix} = \int_{\Omega^-} [B^-(\mathbf{x})]^t [\mathbf{C}^-][B^-(\mathbf{x})]d\Omega \tag{36}$$

where we introduced the strain–displacement matrix $[B^+]$ (resp. $[B^-]$), which contains the derivatives of the shape functions in $\Omega^+$ (resp. $\Omega^-$). Vectors $\{fext^+\}$ and $\{fext^-\}$ in Equation (31) are the external force vectors induced by the external tractions $\mathbf{t_{ext}}$:

$$\{fext^+\} = \int_{\Gamma_t^+} [N^+(\mathbf{x})]^t \mathbf{t_{ext}^+}(\mathbf{x})d\Gamma \qquad \{fext^-\} = \int_{\Gamma_t^-} [N^-(\mathbf{x})]^t \mathbf{t_{ext}^-}(\mathbf{x})d\Gamma \tag{37}$$

Vectors $\{fcoh^+\}$ and $\{fcoh^-\}$ in Equation (31) are the equivalent force vectors induced by the tractions exerted on the strong discontinuity and are given by

$$\left\{ \begin{matrix} fcoh^- \\ fcoh^+ \end{matrix} \right\} = [K^{coh/sec}] \left\{ \begin{matrix} dcoh^- \\ dcoh^+ \end{matrix} \right\} \tag{38}$$

where $[K^{coh/sec}]$ is the secant finite element matrix associated with the cohesive law Equation (25):

$$[K^{coh/sec}] = \begin{bmatrix} K_{11}^{coh/sec} & K_{12}^{coh/sec} \\ K_{21}^{coh/sec} & K_{22}^{coh/sec} \end{bmatrix} = \int_{\Gamma_c} [B^{coh}(\mathbf{x})]^t [\mathbf{D^{coh/sec}}(\mathbf{x})][B^{coh}(\mathbf{x})]d\Gamma \tag{39}$$

In this study, we utilise classical cohesive elements equipped with displacement degrees of freedom only, as discussed in [38]. Although cohesive elements with drilling degrees of freedom do exist and have been reported in [28], their performance has not been thoroughly studied in the literature. Therefore, we have chosen to stick with the classical cohesive element formulation for this investigation. The use of cohesive elements with drilling degrees of freedom will be considered for future studies.

The substitution of Equations (33), (35), (36) and (38) into (31) leads to

$$\begin{bmatrix} L_{11}^- & 0 \\ 0 & L_{11}^+ \end{bmatrix} \begin{Bmatrix} dext^- \\ dext^+ \end{Bmatrix} + \begin{bmatrix} L_{12}^- & 0 \\ 0 & L_{12}^+ \end{bmatrix} \begin{Bmatrix} dcoh^- \\ dcoh^+ \end{Bmatrix} - \begin{Bmatrix} fext^- \\ fext^+ \end{Bmatrix} = \begin{Bmatrix} 0 \\ 0 \end{Bmatrix} \tag{40}$$

$$\begin{bmatrix} L_{21}^- & 0 \\ 0 & L_{21}^+ \end{bmatrix} \begin{Bmatrix} dext^- \\ dext^+ \end{Bmatrix} + \begin{bmatrix} L_{22}^- + K_{11}^{coh/sec} & K_{12}^{coh/sec} \\ K_{21}^{coh/sec} & L_{22}^+ + K_{22}^{coh/sec} \end{bmatrix} \begin{Bmatrix} dcoh^- \\ dcoh^+ \end{Bmatrix} = \begin{Bmatrix} 0 \\ 0 \end{Bmatrix} \tag{41}$$

Equations (40) and (41) represent the AFEM discretised equilibrium equations. The problem to solve can be summed up as follows: find the degrees-of-freedom vectors $\{dcoh^-\}$, $\{dcoh^+\}$, $\{dext^+\}$ and $\{dext^-\}$ knowing $\{fext^+\}$ and $\{fext^-\}$ such that Equations (40) and (41) are satisfied.

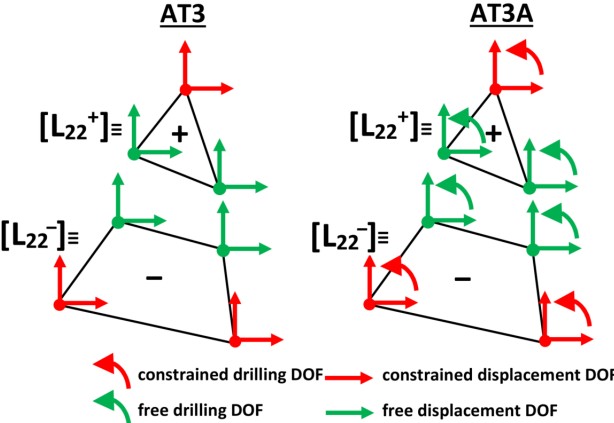

**Figure 7.** Graphical representation of submatrices $[L_{22}^{\pm}]$ of a triangular augmented element, AT3, and a stabilised augmented triangular element, AT3A.

### 3.3. Condensed Discretised Equilibrium Equations and Nested Global/Local Solving Procedure

One critical aspect of augmented finite element methods lies in the nested global/local procedure adopted to solve the coupled equilibrium Equations (40) and (41). Indeed, this procedure allows for the condensation of DOFs related to the strong discontinuity surface, namely, $\{dcoh^+\}$ and $\{dcoh^-\}$. Consequently, cracks can be embedded in elements without requiring additional global DOFs. The condensation process is straightforward: assuming that Equations (40) and (41) have been solved (i.e., their right-hand sides are null), the substitution of Equation (41) into Equation (40) yields:

$$\underbrace{\left( \begin{bmatrix} L_{11}^- & 0 \\ 0 & L_{11}^+ \end{bmatrix} - \begin{bmatrix} L_{12}^- & 0 \\ 0 & L_{12}^+ \end{bmatrix} \begin{bmatrix} L_{22}^- + K_{11}^{coh/sec} & K_{12}^{coh/sec} \\ K_{21}^{coh/sec} & L_{22}^+ + K_{22}^{coh/sec} \end{bmatrix}^{-1} \begin{bmatrix} L_{21}^- & 0 \\ 0 & L_{21}^+ \end{bmatrix} \right)}_{=[K_{AFEM}^{sec}]} \begin{Bmatrix} dext^- \\ dext^+ \end{Bmatrix}$$

$$- \begin{Bmatrix} fext^- \\ fext^+ \end{Bmatrix} = \begin{Bmatrix} 0 \\ 0 \end{Bmatrix} \quad (42)$$

The (secant) stiffness matrix $[K_{AFEM}^{sec}]$ of an element is dependent on the state of the cohesive crack, but it does not explicitly reference the associated degrees of freedom $\{dcoh^{\pm}\}$. Thus, it is commonly referred to as a condensed stiffness matrix, and Equation (42) is known as the condensed discretised equilibrium equation. The matrix-vector products $[K^{coh/sec}].\{dcoh^- \ dcoh^+\}^t$ (Equation (41)) and $[K_{AFEM}^{sec}].\{dext^- dext^+\}^t$ (Equation (42)) are similar to the internal force vector $\int [B]^t \sigma dV$ computed with classical finite elements. In the AFEM framework, the global DOFs associated with cracked and uncracked elements are identical. Therefore, low-level aspects of finite element codes, such as DOF numbering and matrix assembly, are not affected by the use of augmented elements.

Let us now shift our focus to the incremental-iterative strategy adopted to solve Equation (42). In this regard, we introduce indices *n*, *i* and *j* related to increments, local iterations and global iterations, respectively. The problem aims to determine $\{dext^{\pm}\}^{n+1}$, given $\{fext^{\pm}\}^{n+1}$ and $\{dcoh^{\pm}\}^n$ such that Equation (42) holds. The solution to the problem follows a two-stage approach, referred to as the global and local stages, and is elaborated in detail in Algorithm 1. Firstly, we compute a trial value of the external degrees of freedom $\{dext^{\pm}\}^j$ associated with the known load vectors $\{fext^{\pm}\}^{n+1}$, under the assumption that crack openings remain constant, i.e., $\{\Delta d_{coh}^{\pm}\} = \{0\}$. The condensed tangent stiffness matrix of augmented elements is introduced for that purpose and is given by

$$[K_{AFEM}^{tan}] = \begin{bmatrix} L_{11}^- & 0 \\ 0 & L_{11}^+ \end{bmatrix} - \begin{bmatrix} L_{12}^- & 0 \\ 0 & L_{12}^+ \end{bmatrix} \left( \begin{bmatrix} L_{22}^- & 0 \\ 0 & L_{22}^+ \end{bmatrix} + [K^{coh/tan}] \right)^{-1} \begin{bmatrix} L_{21}^- & 0 \\ 0 & L_{21}^+ \end{bmatrix} \quad (43)$$

where $[K^{coh/tan}]$ is the tangent stiffness matrix associated with the cohesive interface, as given in Appendix B. The computation of the trial values of $\{dext^{\pm}\}^j$ does not require the assembly of quantities related to the cohesive interface: vectors $\{dcoh^{\pm}\}^i$ are treated as internal variables. This one-step operation is the global stage.

The local stage consists of finding $\{dcoh^{\pm}\}^i$ such that Equation (41) is satisfied. The external degrees of freedom remain constant during this iterative stage (i.e., $\{\Delta d_{ext}^{\pm}\}^j = \{0\}$). Two markedly different strategies can be adopted during the local stage. The one mostly employed by Yang and coworkers considers cracks as being local to the elements, i.e., cohesive surface-related quantities ($\{dcoh^{\pm}\}$, $[K_{AFEM}^{tan}]$, $[K_{AFEM}^{sec}]$ and $\{fcoh^{\pm}\}$) are not assembled but are instead treated at the element level. Algorithm 1 then becomes similar to a non-linear constitutive law with the difference being that the iterations are performed at the element level and not at the integration point level. These key aspects enable the efficient implementation of augmented elements in any industrial code equipped with user-defined elements because Equation (41) is solved independently within each element. This strategy nevertheless introduces a significant drawback: the interelement compatibility of the displacement field is lost and, to the best of the authors' knowledge, the equivalence between the problem's strong and weak forms remains to be demonstrated. The second possible strategy maintains interelement compatibility of the displacement field. To do so, one follows Algorithm 1 and assembles the cohesive surface-related quantities before solving Equation (41). Because this approach requires DOF management and matrix assembly, it cannot be easily implemented in (closed-source) industrial software. Yang and coworkers coined this second strategy the conforming augmented finite element method (C-AFEM) [40,41].

**Algorithm 1** Calculate $\{dext^\pm\}^{n+1}$ given $\{fext^\pm\}^{n+1}$, $\{dext^\pm\}^n$ and $\{dcoh^\pm\}^n$

---

$\begin{Bmatrix} fext^- \\ fext^+ \end{Bmatrix}^j \leftarrow \begin{Bmatrix} fext^- \\ fext^+ \end{Bmatrix}^{n+1} \qquad \begin{Bmatrix} dext^- \\ dext^+ \end{Bmatrix}^j \leftarrow \begin{Bmatrix} dext^- \\ dext^+ \end{Bmatrix}^n \qquad \begin{Bmatrix} dcoh^- \\ dcoh^+ \end{Bmatrix}^i \leftarrow \begin{Bmatrix} dcoh^- \\ dcoh^+ \end{Bmatrix}^n$

**while** $\left\| \{res_{global}\}^j \right\|_{L2} = \left\| \begin{Bmatrix} fext^- \\ fext^+ \end{Bmatrix}^j - \left[ K_{AFEM}^{sec}\left(\{dcoh^\pm\}^i\right) \right] \begin{Bmatrix} dext^- \\ dext^+ \end{Bmatrix}^j \right\|_{L2} > \epsilon_{global}$ **do**

$\qquad \begin{Bmatrix} \Delta dext^- \\ \Delta dext^+ \end{Bmatrix}^j \leftarrow \left[ K_{AFEM}^{tan}\left(\{dcoh^\pm\}^i\right) \right]^{-1} \{res_{global}\}^j$

$\qquad \begin{Bmatrix} dext^- \\ dext^+ \end{Bmatrix}^j \leftarrow \begin{Bmatrix} dext^- \\ dext^+ \end{Bmatrix}^j + \begin{Bmatrix} \Delta dext^- \\ \Delta dext^+ \end{Bmatrix}^j$

$\qquad$ **while** $\left\| \{res_{local}\}^i \right\|_{L2} = \left\| \begin{bmatrix} L_{21}^- & 0 \\ 0 & L_{21}^+ \end{bmatrix} \begin{Bmatrix} dext^- \\ dext^+ \end{Bmatrix}^j + \left( \begin{bmatrix} L_{22}^- & 0 \\ 0 & L_{22}^+ \end{bmatrix} + \left[ K^{coh/sec}\left(\{dcoh^\pm\}^i\right) \right] \right) \begin{Bmatrix} dcoh^- \\ dcoh^+ \end{Bmatrix}^i \right\|_{L2} >$

$\epsilon_{local}$ **do**

$\qquad\qquad \begin{Bmatrix} \Delta dcoh^- \\ \Delta dcoh^+ \end{Bmatrix}^i \leftarrow \left( \begin{bmatrix} L_{22}^- & 0 \\ 0 & L_{22}^+ \end{bmatrix} + \left[ K^{coh/tan}\left(\{dcoh^\pm\}^i\right) \right] \right)^{-1} \{res_{local}\}^i$

$\qquad\qquad \begin{Bmatrix} dcoh^- \\ dcoh^+ \end{Bmatrix}^i \leftarrow \begin{Bmatrix} dcoh^- \\ dcoh^+ \end{Bmatrix}^i + \begin{Bmatrix} \Delta dcoh^- \\ \Delta dcoh^+ \end{Bmatrix}^i$

$\qquad$ **end while**

**end while**

$\begin{Bmatrix} dext^- \\ dext^+ \end{Bmatrix}^{n+1} \leftarrow \begin{Bmatrix} dext^- \\ dext^+ \end{Bmatrix}^j \qquad \begin{Bmatrix} dcoh^- \\ dcoh^+ \end{Bmatrix}^{n+1} \leftarrow \begin{Bmatrix} dcoh^- \\ dcoh^+ \end{Bmatrix}^i$

---

The second (conforming) strategy is theoretically more appealing because it ensures weak- and strong-form equivalence. Still, we will focus on the first (nonconforming) approach to authorise the implementation of the method in industrial software. Furthermore, the nonconforming strategy has been observed to outperform well-established methods such as the FEM or the XFEM in some benchmarks [13,16].

### 3.4. Treatment of Stiffness Matrices Singularities

The inverse of some stiffness submatrices appears in the AFEM discretised equilibrium equation (Equation (42)). The invertibility of these submatrices depends on two factors: (i) the parent element used, such as the constant strain triangular element (T3) or the bilinear quadrilateral element (Q4), and (ii) the crack's location inside the element. Essongue and colleagues have shown that using the T3 parent elements leads to the singularity of these submatrices when cohesive tractions are absent [16]. To help clarify, assume that the cohesive discontinuity is completely damaged and opened, meaning that the values for $d_n$ and $d_t$ in Equation (25) are both equal to 1, and $\delta_n$ in Equation (A3) is greater than 0. This causes the cohesive finite element matrix, defined by Equation (39), to become a null matrix, $[K^{coh/sec}] = [0]$. To use Equation (42), it is necessary for submatrices $[L_{22}^+]$ and $[L_{22}^-]$ to be invertible. However, when adopting the T3 parent elements, $[L_{22}^+]$ becomes singular. To understand the invertibility of submatrices $[L_{22}^\pm]$, it is helpful to consider their mechanical interpretation. $[L_{22}^+]$ (and $[L_{22}^-]$) is the stiffness matrix for subdomain $\Omega^+$ (and $\Omega^-$) with a fully constrained DOF $\{dext^+\}$ (and $\{dext^-\}$) [16]. This is illustrated in Figure 7 for both the triangular augmented element (AT3) and the stabilised augmented triangular element (AT3A).

As can be seen in Figure 7, the upper triangular region of the AT3 is free to rotate, resulting in a singular $[L_{22}^+]$. Drilling DOFs are used to prevent this rigid-body rotation and ensure full rank of $[L_{22}^+]$. The effectiveness of this approach is demonstrated in the numerical experiments presented in Section 4.1.1.

Although the originators of the AFEM noted that "the nonzero stiffness in cohesive law helps to regularize the matrix condition" [6], they did not provide guidance on how to handle a fully softened cohesive zone. To our knowledge, the only published strategy for dealing with this situation is to compute the pseudoinverse of $[L_{22}^+]$, as proposed in our previous work [16]. Our numerical experiments have demonstrated that this method converged with mesh refinement and competed with the FEM or XFEM. Therefore, we use this approach with AT3 elements. The AT3A does not require pseudoinverse computations

because the introduction of drilling DOFs suppresses the occurrence of singular stiffness submatrices (as shown in Section 4.1.1).

The AFEM is not the only embedded crack method that produces singular matrices. Several embedded finite element methods (EFEMs) also share this drawback. However, stabilisation strategies used in EFEMs cannot be directly applied to the AFEM. This is because EFEMs use prescribed crack kinematics (such as a constant or linear crack opening) for stabilisation [42–44], unlike the AFEM, which does not compute a crack opening directly. In the AFEM, the subdomains' displacements are first solved, and then the crack opening is computed (using Equations (34), (41) and (40)). Moreover, the two subdomains of an augmented element have different kinematics (Equation (32)), a feature not shared by most EFEMs [31,32,43]. Therefore, stabilisation procedures that have been developed for EFEMs cannot be directly applied to the AFEM due to their differences.

### 3.5. Working with Various Element Shapes

While the upcoming numerical experiments will primarily concentrate on 2D triangular elements, it is important to note that the SAFEM is applicable in 3D scenarios and with various element shapes. The only prerequisite is the ability to construct the stiffness matrix associated with subdomains $\Omega^+$ and $\Omega^-$ (Equation (42)), which might involve some additional effort. Suppose, for instance, that the initial crack-free elements are Allman quadrilaterals (Q4A). If the Q4A is augmented due to the nucleation of a crack that splits the element into triangular and pentagonal subdomains, the triangular domain is to be modelled with the T3A presented in Section 2. One then needs to derive a pentagonal element with drilling DOFs, which would be termed P5A, to model the pentagonal subdomain. To the best of the authors' knowledge, this element is absent from the literature. Nevertheless, the methodology to derive an Allman element from an underlying high-order element is straightforward (cf. Section 2). Hence, in the present example, one could use the high-order pentagonals proposed in Ref. [45] or [46] to compute the stiffness matrix of the P5A. Importantly, the SAFEM algorithm remains unmodified.

## 4. Numerical Experiments

This section compares the performances of the AFEM and the SAFEM. To allow for detailed observations, the numerical experiments are divided into two sections: traction-free strong discontinuities are investigated in Section 4.1 while cohesive discontinuities are considered in Section 4.2. To prompt the need for stabilisation, the computations will be performed exclusively on 2D triangular meshes. The material and geometric parameters will be given without units as is customary in computational mechanics [18,21,26].

### 4.1. Traction-Free Strong Discontinuities

While the AFEM was primarily designed to simulate the propagation of cohesive cracks, studying its ability to represent traction-free discontinuities is also of paramount importance. Indeed, traction-free situations put to light some AFEM deficiencies that (i) may remain undetected when cohesive tractions are large enough but (ii) could result in the termination of a simulation when interfaces are fully damaged. The next two sections investigate the singularities of the matrices arising in the AFEM formulation, as well as its convergence rate in traction-free situations.

#### 4.1.1. The Stabilising Effects of Drilling Degrees of Freedom

As highlighted in Section 3.4, the stiffness matrices of augmented elements may become singular due to an internal rigid-body rotation (Figure 7). To prevent this, drilling DOFs were introduced into the AFEM framework giving rise to the SAFEM. However, Allman's drilling DOFs are not true rotations (as explained in Section 2.1). Therefore, it is reasonable to question whether they effectively suppress the internal rigid-body rotation that occurs in the AFEM framework. To investigate this, we conducted numerical experiments where we computed the lowest eigenvalue of the submatrix $[L_{22+}]$ of the AT3

and AT3A elements, denoted as $\lambda_{AT3}$ and $\lambda_{AT3A}$, for a large number of geometries. These geometries were generated by varying the coordinates $\{x_1, y_1\}$ of the first node of the triangular domain illustrated in Figure 8.

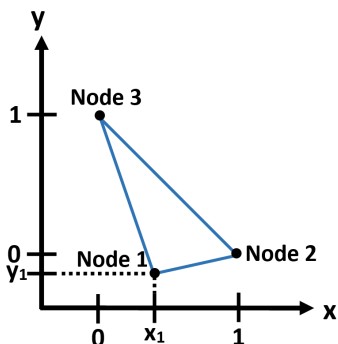

**Figure 8.** Triangular domain employed to compute $[L_{22+}]$ and its eigenvalues. $\{x_1, y_1\}$ are the parametric coordinates of Node 1.

The following isotropic material properties are (arbitrarily) chosen: Young's modulus is $E = 10^7$ and Poisson's ratio is $\nu = 0.3$. The values of $\lambda_{AT3}$ and $\lambda_{AT3A}$ computed for $10^6$ different sets of coordinates $\{x_1, y_1\}$ are plotted in Figure 9.

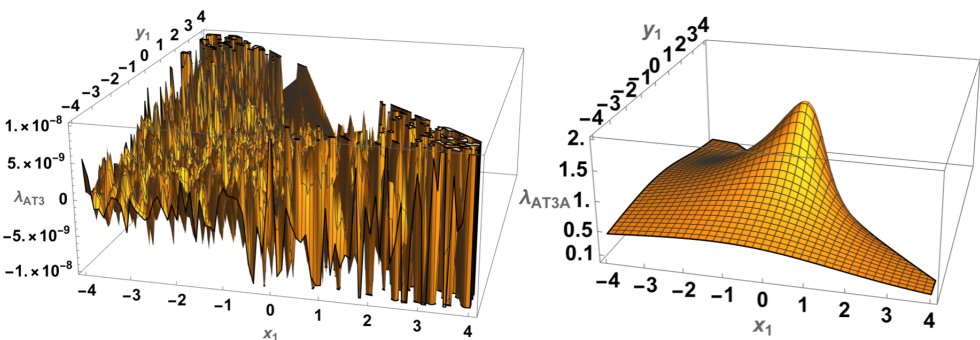

**Figure 9.** Lowest eigenvalues of $[L_{22+}]$ for augmented elements: $\lambda_{AT3}$ (**left**) and $\lambda_{AT3A}$ (**right**) computed for $10^6$ different sets of coordinates $\{x_1, y_1\}$. $\{x_1, y_1\}$ are the coordinates of Node 1, see Figure 8.

$\lambda_{AT3}$ oscillates around 0, illustrating the singularity of $[L_{22+}]$ that occurs when AT3 elements are used. Conversely, $\lambda_{AT3A}$ is always positive and evolves smoothly with the coordinates of Node 1, even in the limited case of a triangular domain collapsed into a line (e.g., $\{x_1, y_1\} = \{1, 0\}$). This is a significant outcome that demonstrates the SAFEM's ability to produce non-singular element stiffness submatrices $[L_{22+}]$ as required by the condensed discretised equilibrium equation (Equation (42)). Consequently, the computation of the pseudoinverse of $[L_{22+}]$, which is necessary in the AFEM, is no longer needed. This simplifies and improves the efficiency of augmented elements because closed-form solutions for the inverse of symmetric matrices can be used advantageously. The next section investigates the impact of drilling DOF incorporation on the accuracy of augmented elements, which is essential to further validate the SAFEM's performance.

### 4.1.2. Convergence of the AFEM and the SAFEM in the Energy and L2 Norms

The goal of this section is to compare the AFEM and the SAFEM convergence behaviour in the energy and L2 norms. To achieve this, we will perform numerical experiments using known analytical solutions. We conduct a test case encountered in the XFEM literature [26,47]: an infinite plate cut by a traction-free and horizontal crack is loaded by a remote stress field and one successively considers the Mode I and Mode II loadings of the

crack. Plane strain is assumed and the material is homogeneous and isotropic with Young's modulus $E$ and Poisson's ratio $\nu$. The analytical expression of the displacement field under Mode I loading, denoted as $\boldsymbol{u_I} = \{u_{Ix}, u_{Iy}\}$, is given by [48]:

$$u_{Ix} = \frac{K_I}{E}\sqrt{\frac{r}{2\pi}}(1+\nu)cos\left(\frac{\theta}{2}\right)(3-4\nu-cos(\theta)) \tag{44}$$

$$u_{Iy} = \frac{K_I}{E}\sqrt{\frac{r}{2\pi}}(1+\nu)sin\left(\frac{\theta}{2}\right)(3-4\nu-cos(\theta)) \tag{45}$$

where $K_I$ is the Mode I stress intensity factor and $(r,\theta)$ are the polar coordinates associated with a reference frame centred at the crack tip as depicted in Figure 10.

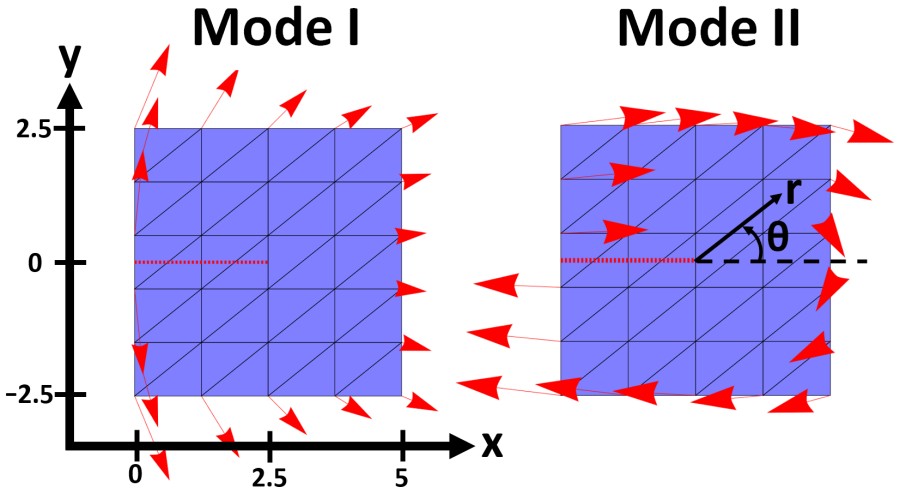

**Figure 10.** Polar and Cartesian coordinate systems, geometry and displacement boundary conditions imposed on the boundary of the plate under Mode I and Mode II loadings of the crack.

The closed-form solution of the displacement field under Mode II loading, denoted as $\boldsymbol{u_{II}}$, is [48]:

$$u_{IIx} = \frac{K_{II}}{E}\sqrt{\frac{r}{2\pi}}(1+\nu)sin\left(\frac{\theta}{2}\right)(5-4\nu+cos(\theta)) \tag{46}$$

$$u_{IIy} = \frac{K_{II}}{E}\sqrt{\frac{r}{2\pi}}(1+\nu)cos\left(\frac{\theta}{2}\right)(-1+4\nu-cos(\theta)) \tag{47}$$

The numerical experiments are performed with a square domain $\Omega = [0,5] \times [-2.5, 2.5]$ cut by a crack $\Gamma_c = [0, 2.5] \times \{0\}$, see Figure 10. Young's modulus and Poisson's ratio are $E = 200,000$ and $\nu = 0.3$, respectively, and the imposed stress intensity factors are $K_I = K_{II} = 2802, 5$. Following Laborde and coworkers [47], we impose the closed-form displacement field on the boundary of the plate, as depicted in Figure 10. Eight homogeneous meshes with element sizes ranging from $h = 0.025$ to $h = 1.25$ are employed to discretise the square domain and compare the relative accuracy of the AFEM and the SAFEM under the Mode I and Mode II loadings of the crack. The expression of the relative error in the energy norm was introduced in Section 2.2.2 (cf. Equation (13)) and is not repeated here. The relative error in the L2-norm is given by:

$$||\mathbf{e}||_{L2} = \left(\frac{\int_\Omega (\boldsymbol{u}-\hat{\boldsymbol{u}})\cdot(\boldsymbol{u}-\hat{\boldsymbol{u}})\,d\Omega}{\int_\Omega \boldsymbol{u}\cdot\boldsymbol{u}\,d\Omega}\right)^{\frac{1}{2}} \tag{48}$$

where $u$ is the exact displacement field ($u = u_I$ in Mode I and $u = u_{II}$ in Mode II) and $\hat{u}$ is the displacement field provided by the chosen numerical method (i.e., the AFEM or the SAFEM).

The energy error of the SAFEM and the AFEM under the Mode I loading of the crack is plotted in Figure 11. One observes that the convergence rates of the AFEM and the SAFEM are identical. This convergence rate value ($||e|| = \mathcal{O}(h^{0.5})$) is well understood: it is caused by the crack-induced singularity of the stress field [49]. One also notices that the SAFEM outperforms the AFEM for a given element size and a given number of DOFs.

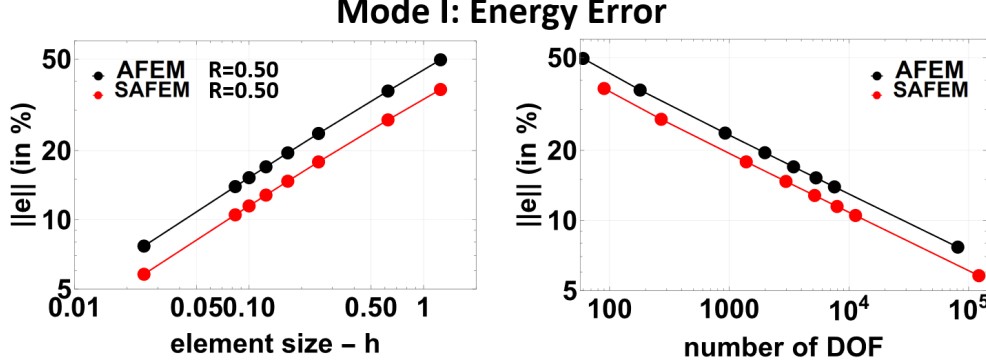

**Figure 11.** Mode I loading of the crack: error in the energy norm $||e||$ as a function of the mesh size $h$ (**left**) or the number of degrees of freedom (**right**) with the AFEM and the SAFEM. R indicates the method convergence rate such that $||e|| = \mathcal{O}(h^R)$.

The error in the energy norm under the Mode II loading of the crack is plotted in Figure 12. The AFEM and the SAFEM share the same convergence rate, $||e|| = \mathcal{O}(h^{0.43})$, and are of similar accuracy. As stated previously, a convergence with an error of $\mathcal{O}(h^{0.5})$ is expected in this situation. Numerical experiments performed in Ref. [29] revealed that the AFEM was overly soft under Mode II loadings, leading to suboptimal convergence rates in the energy norm. The similarities between the two curves plotted in Figure 12 suggest that the SAFEM shares this drawback.

Although the error levels shown in Figures 11 and 12 may appear significant, it is worth noting that such errors are commonly observed when modelling singularities using the FEM or the XFEM. This observation is supported by the comparison between the FEM and the XFEM in Ref. [47] (refer to Figure 9). Likewise, the quantitative comparison between the AFEM and the FEM presented in [29] (refer to Figures 16 and 17) indicates similar levels of error and a virtually identical performance of the AFEM and the FEM under the Mode I loading of a crack. Therefore, the fact that the SAFEM is either as performant as (under Mode II loading) or outperforms the AFEM (under Mode I loading) makes it a highly attractive option for researchers and practitioners in the field.

Let us now consider the error in the L2-norm, also denoted as the displacement error. A convergence with an error of $\mathcal{O}(h)$ is expected in this situation [50]. Figure 13 compares the relative performances of the SAFEM and the AFEM under the Mode I loading of the crack. The differences between the two methods are considerable: the SAFEM converges at $\mathcal{O}(h^{1.32})$, while the AFEM is suboptimal with an $\mathcal{O}(h^{0.52})$ convergence rate. A similar situation is also observed under the Mode II loading of the crack where the SAFEM outperforms the AFEM, see Figure 14. In that case, the SAFEM convergence rate, $\mathcal{O}(h^{0.74})$, is suboptimal but remains higher than the AFEM one, $\mathcal{O}(h^{0.5})$.

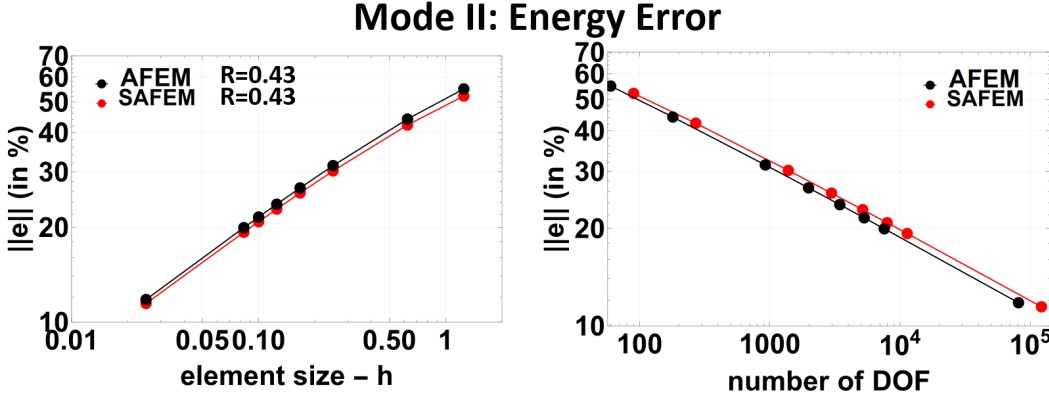

**Figure 12.** Mode II loading of the crack: error in the energy norm $||e||$ as a function of the mesh size $h$ (**left**) or the number of degrees of freedom (**right**) with the AFEM and the SAFEM. R indicates the method convergence rate such that $||e|| = \mathcal{O}(h^R)$.

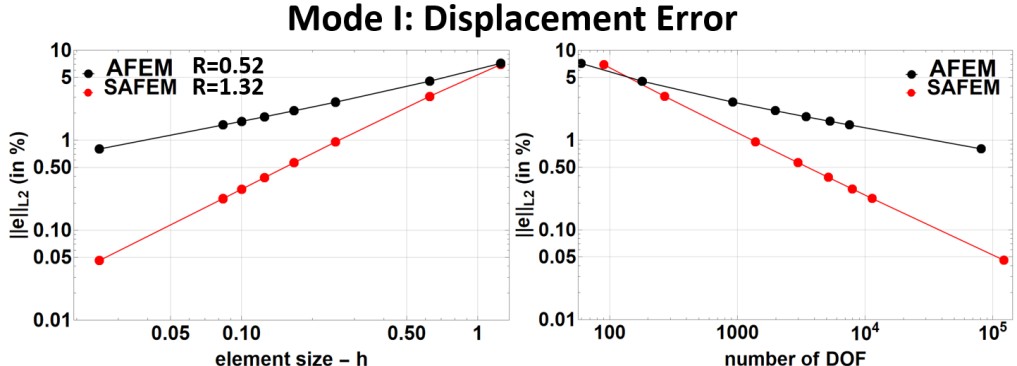

**Figure 13.** Mode I loading of the crack: error in the L2-norm $||e||_{L2}$ as a function of the mesh size $h$ (**left**) or the number of degrees of freedom (**right**) with the AFEM and the SAFEM. R indicates the convergence rate such that $||e|| = \mathcal{O}(h^R)$.

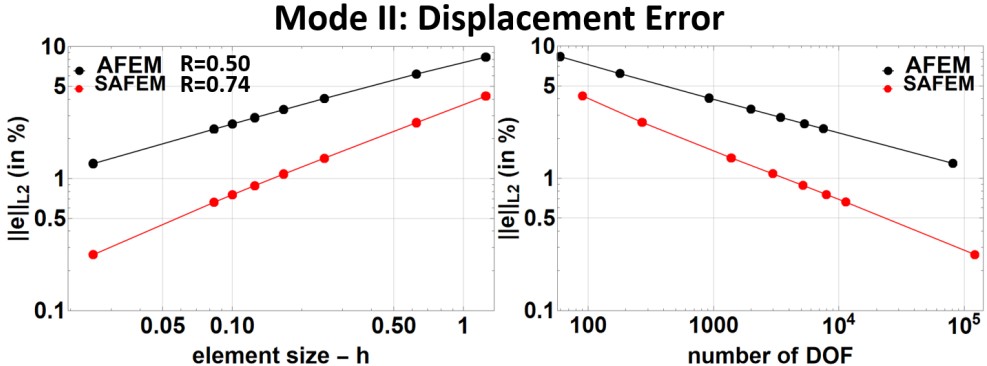

**Figure 14.** Mode II loading of the crack: error in the L2-norm $||e||_{L2}$ as a function of the mesh size $h$ (**left**) or the number of degrees of freedom (**right**) with the AFEM and the SAFEM. R indicates the method convergence rate such that $||e|| = \mathcal{O}(h^R)$.

Additional post-treatments shed light on the underperformance of the AFEM in the L2-norm. To allow for a detailed comparison between the AFEM and the SAFEM, the displacement error at the element level under the Mode I loading of the crack is plotted in Figure 15. This plot reveals that the displacement error is not governed by the crack-tip singularity but by the incompatibility of the interelement displacement field, inherent to the use of augmented elements. The SAFEM reduces this incompatibility, which explains its better accuracy.

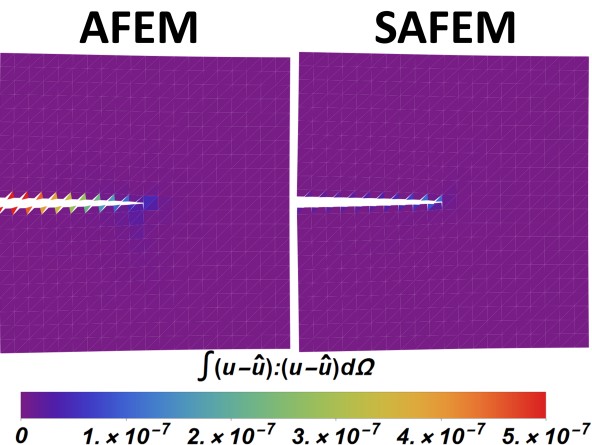

**Figure 15.** Comparison of the AFEM (**left**) and the SAFEM (**right**) displacement error at the element level, under Mode I loading of the crack, with a mesh size $h = 0.25$ (displacements are magnified $3\times$). $u$ and $\hat{u}$ are the exact and the approximate displacement fields, respectively.

This aspect is quantitatively illustrated in Figure 16 where one compares the exact and the approximate normal opening of the crack under Mode I loading. The normal opening of the crack, $\Delta u_n(r)$, is defined as:

$$\Delta u_n(r) = u_{Iy}(r, \theta = \pi) - u_{Iy}(r, \theta = -\pi) \ r \in [0, 2.5] \tag{49}$$

where $u_{Iy}$ is the y-component of the closed-form displacement field, see Equation (45).

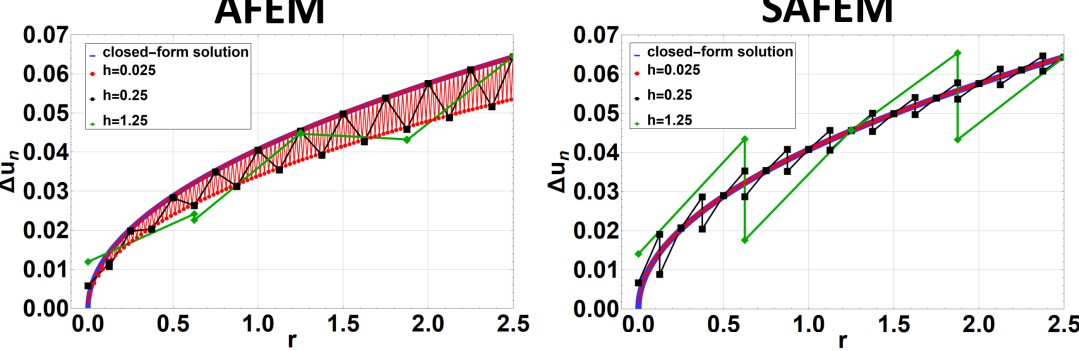

**Figure 16.** Normal opening of the crack under Mode I loading, $\Delta u_n$, computed with the AFEM (**left**) and the SAFEM (**right**) and mesh sizes $h = 0.025$, $h = 0.25$ and $h = 1.25$. $r$ is the polar coordinate along the crack lips, cf. Figure 10.

As shown in Figure 16, (i) the crack opening computed with the AFEM oscillates strongly, (ii) these oscillations do not significantly decrease with mesh refinement and (iii) the crack openings do not fluctuate around the exact values. In contrast, the crack openings computed with the SAFEM consistently converge toward the exact values. The same trends are observed under the Mode II loading of the crack (not shown for the sake of brevity).

The SAFEM and the AFEM exhibit similar performances in terms of the errors in the energy norm, with a slight advantage for the SAFEM. This error measure is dominated by the singularity of the stress field at the crack tip (cf. [49] or Figure 18 of Ref. [16]), where standard finite elements, specifically the T3 or the T3A, are used. The T3 and the T3A have identical convergence rates in the energy norm (Figure 5), which explains the similar performances of the AFEM and the SAFEM in this norm. However, when examining the displacement error, it becomes apparent that the interelement discontinuities of the augmented elements, namely, the AT3 or the AT3A, dominate the error (Figure 15). The

SAFEM predicts the crack opening much more accurately than the AFEM (Figure 16). The stabilisation brought by the drilling DOFs is thus responsible for the superior performance of the SAFEM over the AFEM.

Despite the increased problem size caused by the use of drilling DOFs, the improved accuracy of the SAFEM more than compensates for it. Therefore, it can be concluded that the SAFEM is superior to the AFEM in modelling traction-free cracks for the reasons stated in Table 2.

**Table 2.** Benefits and drawbacks of the SAFEM over the AFEM when modelling traction-free cracks.

| Pros | Cons |
|---|---|
| slightly more accurate in the energy norm, under Mode I loading of a crack (Figure 11) faster convergence in the L2-norm (Figures 13 and 14) crack openings converge toward expected values (Figure 16) | additional implementation effort required due to the use of Allman's elements |

*4.2. Cohesive Strong Discontinuities*

The AFEM was originally developed for cohesive crack propagation [6]. In this section, we shift our focus to cohesive crack modelling and present two sets of tests to compare the performance of the AFEM and SAFEM. The first set of tests examines the satisfaction of a discontinuous patch test, detailed in Section 4.2.1. The second set involves a delamination test case, discussed in Section 4.2.2.

4.2.1. A Discontinuous Patch Test

Patch tests are numerical problems that have exact solutions that lie within the span of a chosen numerical method. Passing patch tests requires that the numerical results match the exact solution. Such tests are often performed during the design of new finite elements as they may provide necessary and sufficient requirements for convergence (cf. Chap. 3 of Ref. [5]). To the best of our knowledge, it has not yet been demonstrated whether passing or failing a patch test ensures or prevents the convergence of the numerical methods that deal with embedded cohesive cracks, such as the AFEM. Therefore, the patch test presented in this section should not be considered as a means to rigorously study the convergence of the AFEM, but rather as a way to highlight its behaviour in simple situations. The discontinuous patch test was first proposed by Armero and coworkers [43] and subsequently adopted by several authors working on finite elements with embedded strong discontinuities [44,51]. The structure considered for the patch test is the $1 \times 1$ isotropic block depicted in Figure 17. It is initially discretised with crack-free standard finite elements, the T3 and the T3A, and subjected to displacement boundary conditions inducing a uniaxial stress state in the entire domain. The displacement boundary conditions shown in Figure 17 are given by:

$$-u(x = 0) = u(x = 1) = u_x \tag{50}$$

$$v(x = 0, y = 0) = v(x = 1, y = 0) = 0 \tag{51}$$

where $u$ and $v$ are the horizontal and vertical displacement fields, respectively. Once the material critical tensile stress $\sigma^c$ is reached, a cohesive discontinuity is inserted through the block's height, dividing it into two equally sized parts. The AFEM or SAFEM are utilised to embed the discontinuity within the finite elements in the numerical experiments. The structure undergoes progressive softening and eventually becomes stress-free as the crack opens. The loading direction is then reversed leading to the closure of the crack and a compressive stress state.

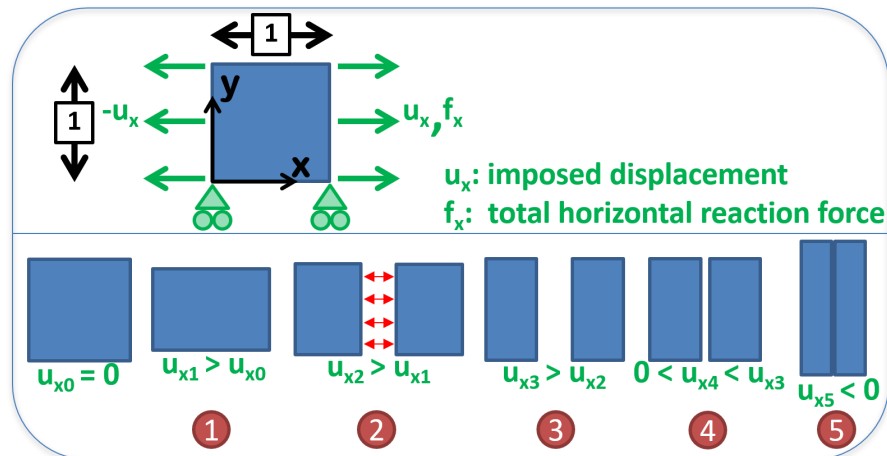

**Figure 17.** Structure geometry, boundary conditions (**top**) and loading scenario through the discontinuous patch test (**bottom**). Stage 1: elastic crack-free structure, Stage 2: structure cut by a softening cohesive discontinuity, Stages 3–4: structure cut by an open and traction-free crack and Stage 5: crack closure. $u_{xi}$ with $1 \leq i \leq 5$ represents the values of the imposed displacement during Stage $i$.

Plane stress conditions are assumed. The bulk material properties are as follows: Young's modulus is $E = 1000$ and Poisson's ratio is $\nu = 0.2$. A bilinear weakly coupled cohesive law (cf., Figure A1) further detailed in Appendix B is employed with the following material properties: the normal and tangential critical tractions are $\sigma^c = 100$ and $\tau^c = 10$, the normal and tangential critical openings are $\delta_n^c = \delta_s^c = 0.001$ and the normal and tangential final openings are $\delta_n^f = \delta_s^f = 0.3$.

The closed-form solution for this test, which demonstrates that the block experiences a uniform and uniaxial stress state $\sigma_{xx}$, is provided in Ref. [43]. We introduce an equivalent stress, $\overline{\sigma_{xx}}$, to assess the distance between the numerical and exact solutions. It is computed thanks to the horizontal reaction force, $f_x$ (cf. Figure 17), measured through the numerical experiment and given by $\overline{\sigma_{xx}} = f_x/S_x$, where $S_x$ is the surface of the block normal to the loading direction (equal to 1 in the present case). This equivalent stress makes it easier to compare the numerical and exact solutions as well as to analyse the influence of mesh refinement. If the numerical method matches the exact solution, $\overline{\sigma_{xx}} = \sigma_{xx}$, and the h-convergence of the numerical method is simply represented by the curves $\overline{\sigma_{xx}}$ and $\sigma_{xx}$ getting closer. Furthermore, because the evaluation of $\overline{\sigma_{xx}}$ involves computing reaction forces, it provides access to the total energy dissipated by the structure, a key quantity in cohesive crack modelling.

Another crucial aspect of cohesive crack modelling is the integration scheme of the cohesive forces (Equations (29) and (30)). This is a widely debated topic, with varying suggestions found in the literature (see, e.g., [38,52–55]). To study the influence of the integration scheme of the cohesive forces, we employed seven integration schemes (ISs), described in Chap. 5.9 of Ref. [5], including one-point ISs; two-, three- and four-point Gauss ISs; and two-, three- and four-point Newton–Cotes ISs.

Three meshes made of randomly oriented triangular elements have been used to study the influence of spatial discretisations on the results (Figure 18). The intersections between the strong discontinuity surface and the triangular elements are arbitrarily close from the mesh nodes.

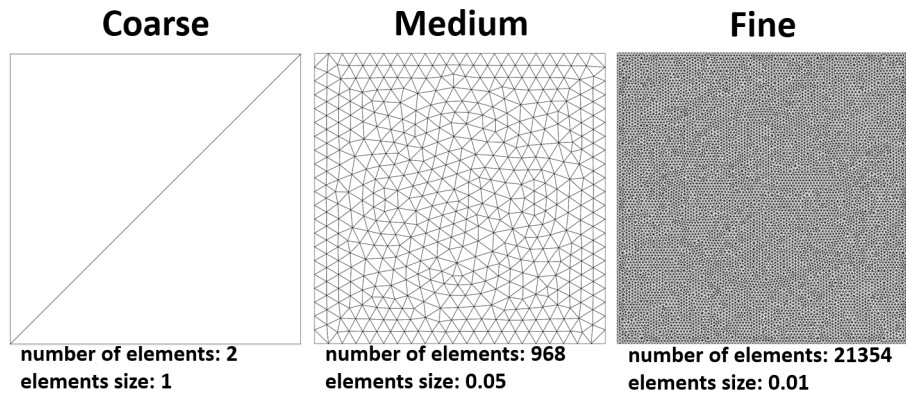

**Figure 18.** Spatial discretisations employed to conduct the discontinuous patch test.

In Figure 19, we present the results obtained using the coarsest spatial discretisation and one-point as well as Gauss integration schemes. Some notable observations can be made regarding both the AFEM and the SAFEM. Firstly, they fail the patch test with all the tested integration schemes. Secondly, a sharp stress drop occurs once the strong discontinuity is introduced. Thirdly, the compressive stress state is not accurately simulated due to the interelement discontinuities inherent in the use of augmented elements. Instead of exerting pressure on the crack lips, the triangular subdomains rotate, as shown in Figure 19. This flaw was found to gradually disappear with mesh refinement. Still, we recommend using Newton–Cotes integration schemes, which induce nonzero compressive stresses, as demonstrated in Figure 20. Hence, we will focus on Newton–Cotes integration schemes in the subsequent analysis.

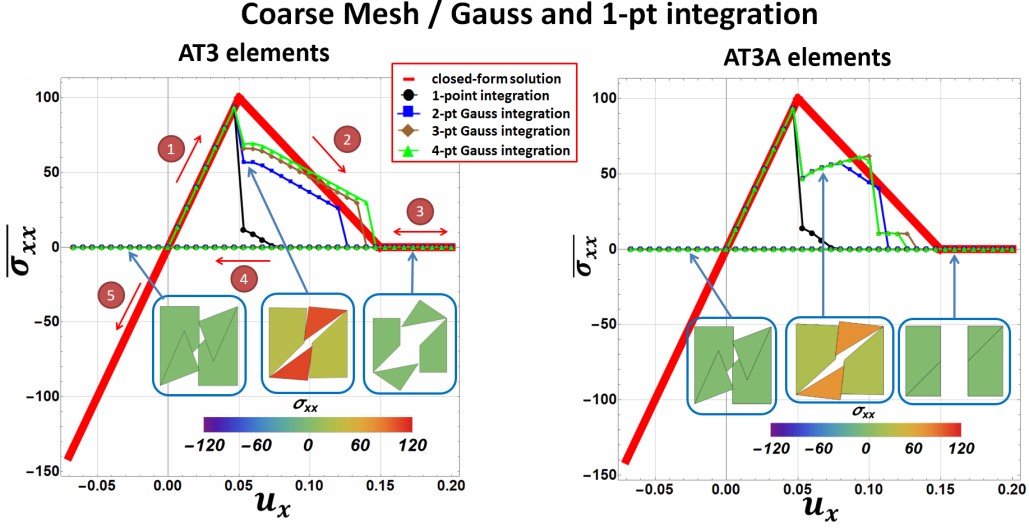

**Figure 19.** Evolution of the equivalent stress $\overline{\sigma_{xx}}$ as a function of the imposed displacement $u_x$ obtained with the AFEM (AT3 elements) and the SAFEM (AT3A elements) for the coarse mesh of this study. The computed stress field $\sigma_{xx}$ is plotted at several steps of the loading scenario. Integration of the cohesive forces: 1-point and 2-, 3- and 4-point Gauss schemes.

The comparison between the AFEM and the SAFEM reveals some differences that hold for all the integration schemes (ISs), as shown in Figures 19 and 20. These differences include (i) the SAFEM exhibiting a softer response in compression than the AFEM, (ii) the SAFEM leading to larger stress drops than the AFEM and (iii) the AFEM producing a non-uniform crack opening under traction-free conditions, which is in stark contrast with the expected constant opening (see Step 3 of Figure 17).

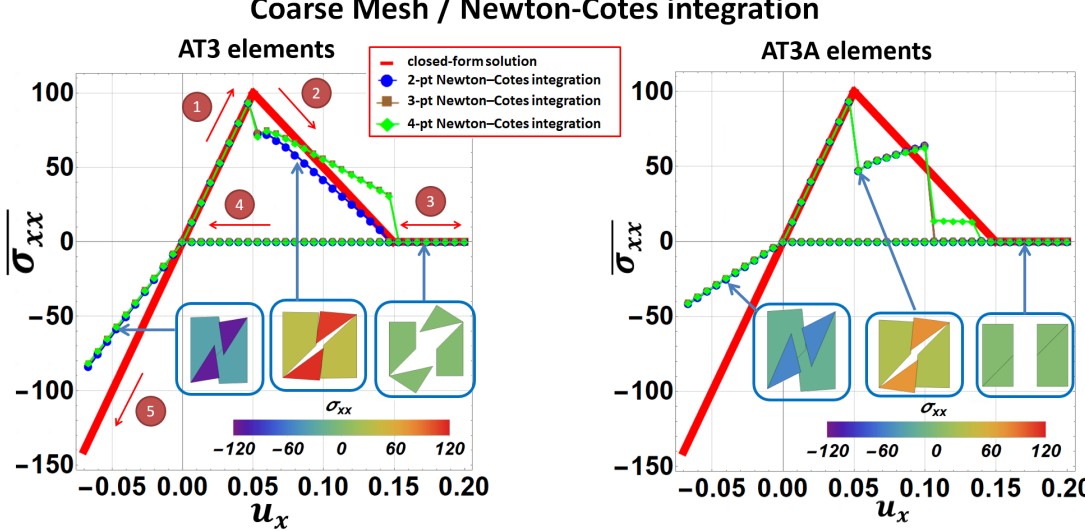

**Figure 20.** Evolution of the equivalent stress $\overline{\sigma_{xx}}$ as a function of the imposed displacement $u_x$ obtained with the AFEM (AT3 elements) and the SAFEM (AT3A elements) for the coarse mesh of this study. The computed stress field $\sigma_{xx}$ is plotted at several steps of the loading scenario. Integration of the cohesive forces: 2-, 3- and 4-point Newton–Cotes schemes.

It is worth noting that flaws (i) and (ii) disappear with mesh refinement, as shown in Figures 21 and 22. However, erroneous crack openings computed with the AFEM are not corrected by finer discretisations as illustrated in Figure 21. Therefore, once again, the SAFEM exhibits superior performance over the AFEM.

## Medium Mesh / Newton-Cotes integration

### AT3 elements

### AT3A elements

**Figure 21.** Evolution of the equivalent stress $\overline{\sigma_{xx}}$ as a function of the imposed displacement $u_x$ obtained with the AFEM and the SAFEM for the medium mesh of this study. The computed stress field $\sigma_{xx}$ is plotted at several steps of the loading scenario. Integration of the cohesive forces: 2-, 3- and 4-point Newton–Cotes schemes.

Furthermore, our tests demonstrate that the three-point Newton–Cotes IS provides optimal results in this test case, as it is both similar to the four-point Newton–Cotes IS and more accurate than the two-point Newton–Cotes IS. We thus chose the three-point Newton–Cotes scheme to visualise the impact of mesh refinement on the results in Figure 22. Interestingly, we found that further mesh refinement does not improve the solutions

significantly, as medium and fine meshes yield similar results. This conclusion holds for other integration schemes as well.

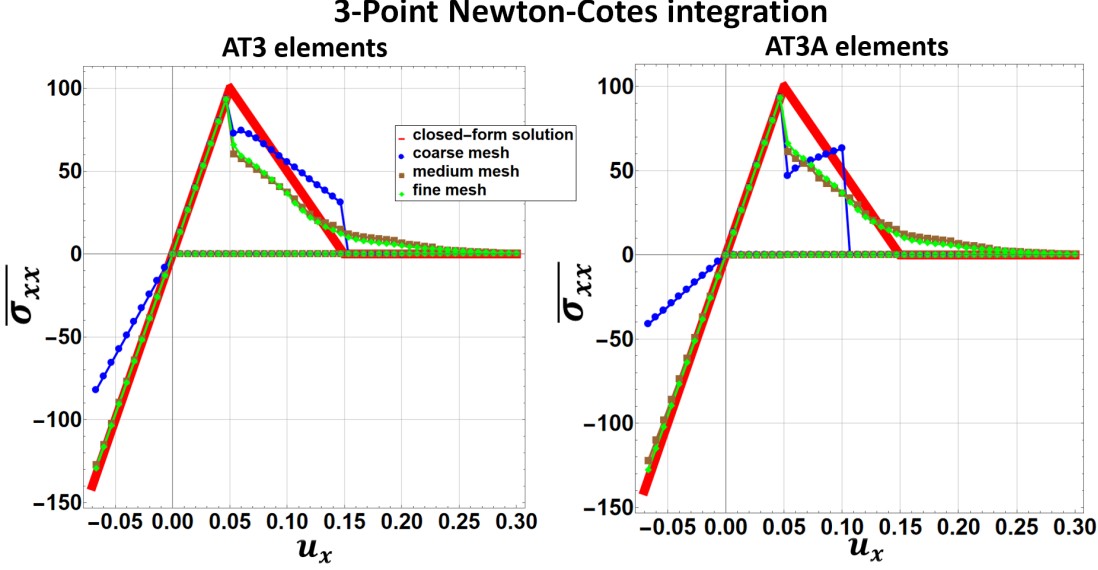

**Figure 22.** Evolution of the equivalent stress $\overline{\sigma_{xx}}$ as a function of the imposed displacement $u_x$ with all the meshes and the 3-point Newton–Cotes integration scheme.

To evaluate the discrepancy between the exact and numerical solutions quantitatively, we chose to compare the dissipated energy. The exact dissipated energy is the product of the cohesive surface fracture energy under the Mode I loading $G_I^c$ (as given by Equation (A6)) and the crack surface area $S_x$, which is equal to 1 in this specific case. Thus, $E_{exact} = G_I^c \times S_x$. The dissipated energies associated with the numerical experiments are computed as $E_{numerical} = 2 \times \int_{u=0}^{max(u_x)} f_x(u)du$, where $f_x$ is the horizontal reaction force and $max(u_x)$ is the maximum imposed displacement. The signed relative error in the dissipated energy, denoted as $E_\%$, is plotted in Figure 23. This graphic illustrates the superior performances of the three- and four-point Newton–Cotes ISs. It also confirms that neither the AFEM nor the SAFEM converge toward the exact solution with mesh refinement.

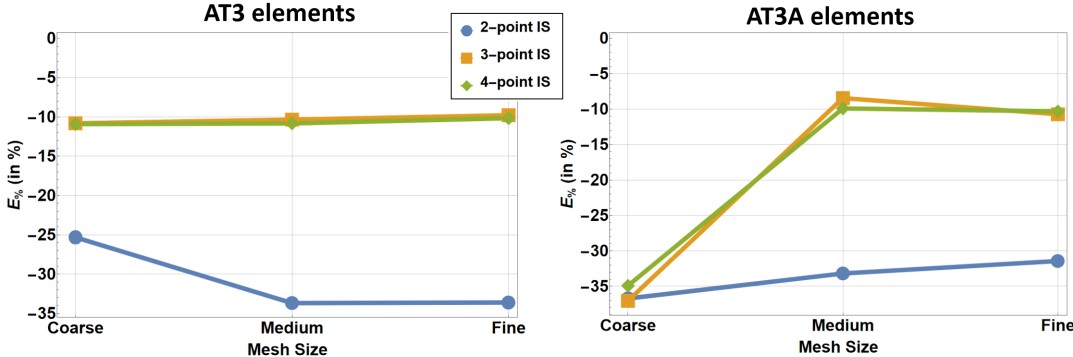

**Figure 23.** Signed relative error in dissipated energy $E_\%$ as a function of mesh refinement with the AFEM (**left**) and the SAFEM (**right**). Two-, three- and four-point Newton–Cotes integration schemes are employed.

### 4.2.2. Mode I Delamination Test

The previous section has highlighted the possible discrepancies between the computed and expected dissipated energies that may occur when augmented elements are made use of. The present section investigates the impact of this flaw on the force/displacement curves obtained in simulations involving multiaxial stress states. A test case with a known analytical solution [56] has been chosen for that purpose: the double-cantilever beam (DCB), see Figure 24. The DCB is used to simulate delamination propagation under pure Mode I loadings. The simulations exactly replicate those of Turon and coworkers [56]. Plane stress conditions are assumed and orthotropic material properties representative of carbon-fiber-reinforced epoxy composites are used: the longitudinal Young's modulus is $E_{xx} = 120$ GPa, Young's modulus in the transverse direction is $E_{yy} = 10.5$ GPa, the in-plane shear modulus is $G_{xy} = 5.25$ GPa and the in-plane Poisson's ratios are $\nu_{xy} = \nu_{yx} = 0.3$. As in Ref. [56], the simulations are performed under displacement control with no specific path-following strategy nor inertia-based stabilisation [8].

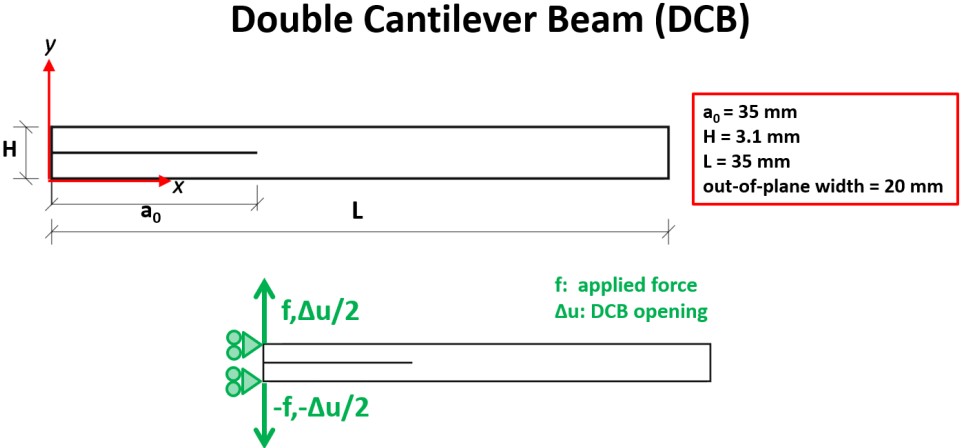

**Figure 24.** Double-cantilever beam (DCB) test specimen: geometry and boundary conditions.

Delamination modelling throughout the DCB test is usually performed with a priori inserted cohesive elements located on the expected delamination plane. The use of augmented elements removes this meshing burden because strong discontinuity surfaces are embedded within the elements. The specimen's initial crack (of coordinates $\{x, y\} \in [0, a_0] \times \{H/2\}$) as well as the expected delamination plane ($\{x, y\} \in [a_0, L] \times \{H/2\}$) are not explicitly meshed but modelled with augmented elements. A traction-free discontinuity is used to represent the initial crack, while the delamination plane is modelled with a weakly coupled bilinear cohesive law (cf. Figure A1), further detailed in Appendix B, and the material properties taken from Ref. [56]. The Mode I and Mode II fracture energies are $G_I^c = 0.26$ N/mm and $G_{II}^c = 1.002$ N/mm, the normal and tangential elastic stiffnesses of the cohesive surface are $K_n = K_s = 10^6$ N/mm$^3$ and the normal and tangential critical stresses are $\sigma^c = 30$ MPa and $\tau^c = 120$ MPa. The cohesive forces are integrated with a three-point Newton–Cotes IS. Three levels of mesh refinement (coarse, medium and fine) are considered in the numerical experiments (Figure 25). The intersections between the strong discontinuity surface and the triangular elements are arbitrarily close from the mesh nodes.

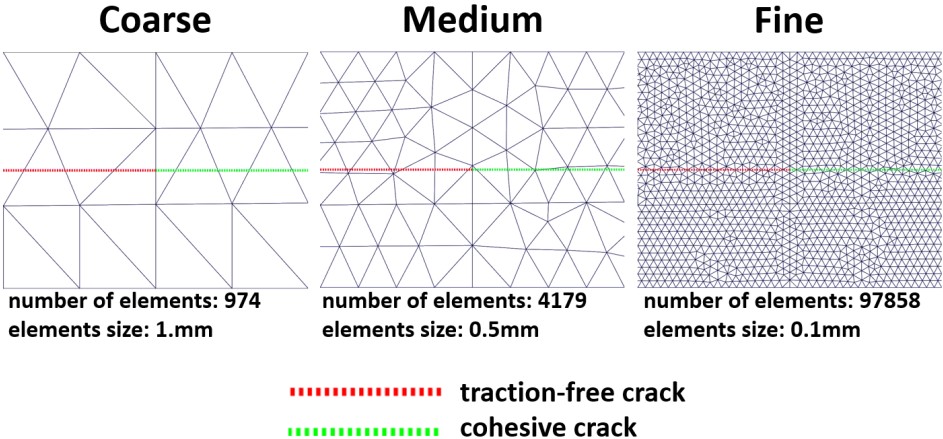

**Figure 25.** Coarse, medium and fine meshes used to run the double-cantilever beam and the end-notch flexure tests. Only the vicinity of the beam's initial crack tip is represented: $\{x, y\} \in [33\,\text{mm}, 37\,\text{mm}] \times [0\,\text{mm}, 3.1\,\text{mm}]$.

Figure 26 compares the numerical and exact force/opening curves obtained when running the DCB test. The observed oscillations are typical in such simulations and tend to disappear with mesh refinement [52,54]. The SAFEM is less prone to oscillations than the AFEM in the present case. The overall results provided by the AFEM and the SAFEM are similar, and both methods converge toward the exact solution with mesh refinement. The convergence is nevertheless deemed "slow". Indeed, the use of mesh-conforming cohesive elements with a 0.3 mm Q4 provides force/opening curves that are closer to the analytical solution than those obtained in this study with the 0.1 mm augmented elements [56].

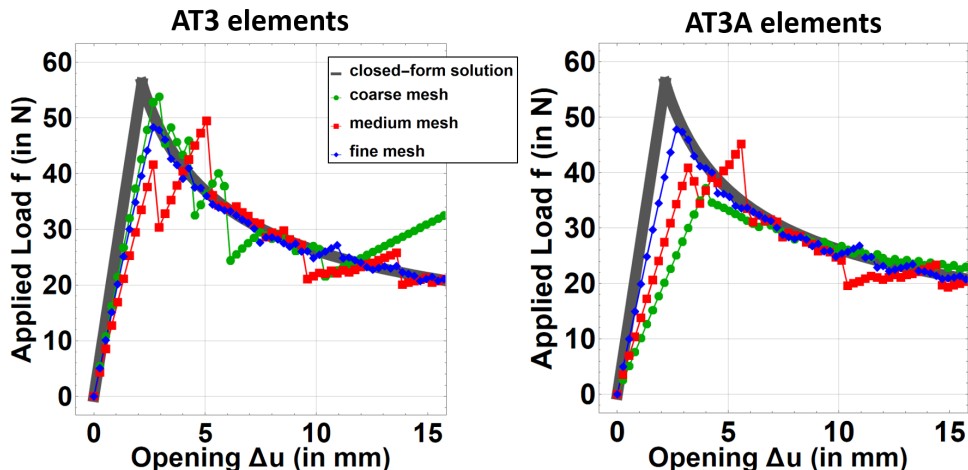

**Figure 26.** DCB test: force ($f$)–opening ($\Delta u$) curves obtained with the AFEM (**left**) and the SAFEM (**right**) with all the meshes of this study.

The reason behind this slow convergence can be grasped by an inspection of the longitudinal stress field $\sigma_{xx}$. As shown in Figure 27, $\sigma_{xx}$ is too low within the augmented elements: they should support the highest longitudinal stress, due to the bending of the two arms, but interelement discontinuities limit the amount of transmitted stresses. This is in line with the observations made in Ref. [29] and highlights the overly soft behaviour of the augmented elements when the crack lips are loaded. Figure 27 puts to light the advantages of augmented elements with drilling DOFs: their use results in a consistent opening of the crack. In contrast, the crack opening computed with the AFEM is inaccurate.

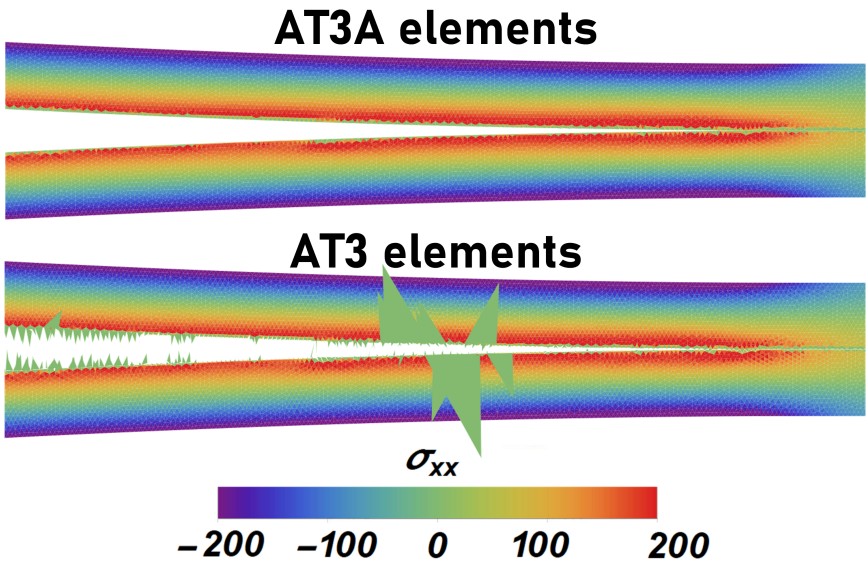

**Figure 27.** DCB test: Longitudinal stress field $\sigma_{xx}$ at the vinity of the crack tip computed with the SAFEM (**top**) and the AFEM (**bottom**) with the finest mesh of this study.

The initial stiffness of the AFEM simulations (represented by the slope of the force/opening curves in Figure 26) remains close to the expected value, even with the coarsest mesh. In contrast, the SAFEM produces an overly soft result with the coarsest mesh. The reason behind this behaviour is straightforward: the AT3 is too soft due to insufficient transmitted stresses (Figure 27), while the T3 element is too stiff (cf. Table 1). These two negative aspects compensate well in the current situation. The T3A is not overly stiff (cf. Table 1), while the AT3A is too soft for the same reason as the AT3. As a result, SAFEM structures are softer than those modelled with the AFEM. The differences disappear with mesh refinement. The benefits and the drawbacks of the SAFEM over the AFEM when modelling cohesive cracks are gathered in Table 3.

**Table 3.** Benefits and drawbacks of the SAFEM over the AFEM when modelling cohesive cracks.

| Pros | Cons |
|------|------|
| smoother force/displacement results in DCB test (Figure 26) consistent crack openings (Figures 21 and 27) | additional implementation effort required due to the use of Allman's elements |

## 5. Conclusions

The AFEM is a validated method that has proven successful in modelling dynamic crack propagation or conducting three-dimensional studies of damage in composite materials. As well known, some combinations of augmented element geometry and crack orientation give rise to singular stiffness matrices when modelling traction-free cracks. This work successfully introduced a method to suppress these singularities, named the stabilised augmented finite element method (SAFEM). The SAFEM relies on the use of bulk elements possessing rotational DOFs. The so-called Allman elements with drilling DOFs have been used for that purpose.

Numerical experiments were conducted to compare the performance of the SAFEM with the traditional AFEM in modelling traction-free cracks and cohesive discontinuities in Mode I and Mode II. The results showed that the AFEM suffers from degraded convergence in the L2-norm and can produce erroneous (cf. Figure 16) and irregular (cf. Figure 27) crack openings. These limitations restrict the applicability of the AFEM in situations where accurate crack opening estimation is essential, such as chemo-mechanical couplings in ceramic matrix composites [57,58]. In contrast, the proposed SAFEM successfully addresses

these issues, limiting interelement discontinuities and achieving an accurate estimation of crack openings when modelling traction-free cracks (Figure 16). The improved accuracy of the SAFEM compensates for the computational cost incurred by the introduction of drilling DOFs (cf. Figures 13 and 14).

The SAFEM's superiority over the AFEM is less pronounced when it comes to modelling cohesive discontinuities. The presence of interelement discontinuities can lead to a premature failure of the cohesive surface, which negatively impacts the accuracy of both the AFEM and the SAFEM. Although both methods have their merits, there is still room for improvement in achieving completely satisfactory results in this area.

This study has demonstrated, for the first time, that even the most straightforward drilling DOF formulation can noticeably enhance the AFEM performance. In future research, efforts will be made to develop tailored drilling DOFs to reduce interelement discontinuities when dealing with cohesive discontinuities.

**Author Contributions:** Conceptualisation, S.E.; methodology, S.E., G.C. and E.M.; software, S.E.; validation, S.E., G.C. and E.M.; formal analysis, S.E., G.C. and E.M.; investigation, S.E. and G.C.; data curation, S.E.; writing—original draft preparation, S.E., G.C. and E.M.; visualisation, S.E.; supervision, G.C. and E.M.; project administration, G.C. and E.M. All authors have read and agreed to the published version of the manuscript.

**Funding:** This research received no external funding.

**Institutional Review Board Statement:** Not applicable.

**Informed Consent Statement:** Not applicable.

**Data Availability Statement:** The data presented in this study are available in the article.

**Conflicts of Interest:** The authors declare no conflict of interest.

## Appendix A. Transformation Matrices

The closed-form expressions of the transformation matrices of two Allman elements, the T3A and the Q4A, are provided hereafter. They are taken from Refs. [1,3].

$$
[T_{\text{T6-T3A}}] =
\begin{bmatrix}
1/2 & 0 & (y_1 - y_2)/8 & 1/2 & 0 & (y_2 - y_1)/8 & 0 & 0 & 0 \\
0 & 1/2 & (x_2 - x_1)/8 & 0 & 1/2 & (x_1 - x_2)/8 & 0 & 0 & 0 \\
0 & 0 & 0 & 1/2 & 0 & (y_2 - y_3)/8 & 1/2 & 0 & (y_3 - y_2)/8 \\
0 & 0 & 0 & 0 & 1/2 & (x_3 - x_2)/8 & 0 & 1/2 & (x_2 - x_3)/8 \\
1/2 & 0 & (y_1 - y_3)/8 & 0 & 0 & 0 & 1/2 & 0 & (y_3 - y_1)/8 \\
0 & 1/2 & (x_3 - x_1)/8 & 0 & 0 & 0 & 0 & 1/2 & (x_1 - x_3)/8
\end{bmatrix}
\tag{A1}
$$

$$
[T_{\text{Q8-Q4A}}] =
\begin{bmatrix}
1 & 0 & 0 & 0 & 0 & 0 & 0 & 0 & 0 & 0 & 0 & 0 \\
0 & 1 & 0 & 0 & 0 & 0 & 0 & 0 & 0 & 0 & 0 & 0 \\
0 & 0 & 0 & 1 & 0 & 0 & 0 & 0 & 0 & 0 & 0 & 0 \\
0 & 0 & 0 & 0 & 1 & 0 & 0 & 0 & 0 & 0 & 0 & 0 \\
0 & 0 & 0 & 0 & 0 & 0 & 1 & 0 & 0 & 0 & 0 & 0 \\
0 & 0 & 0 & 0 & 0 & 0 & 0 & 1 & 0 & 0 & 0 & 0 \\
0 & 0 & 0 & 0 & 0 & 0 & 0 & 0 & 0 & 1 & 0 & 0 \\
0 & 0 & 0 & 0 & 0 & 0 & 0 & 0 & 0 & 0 & 1 & 0 \\
1/2 & 0 & (y1 - y2)/8 & 1/2 & 0 & (y2 - y1)/8 & 0 & 0 & 0 & 0 & 0 & 0 \\
0 & 1/2 & (x2 - x1)/8 & 0 & 1/2 & (x1 - x2)/8 & 0 & 0 & 0 & 0 & 0 & 0 \\
0 & 0 & 0 & 1/2 & 0 & (y2 - y3)/8 & 1/2 & 0 & (y3 - y2)/8 & 0 & 0 & 0 \\
0 & 0 & 0 & 0 & 1/2 & (x3 - x2)/8 & 0 & 1/2 & (x2 - x3)/8 & 0 & 0 & 0 \\
0 & 0 & 0 & 0 & 0 & 0 & 1/2 & 0 & (y3 - y4)/8 & 1/2 & 0 & (y4 - y3)/8 \\
0 & 0 & 0 & 0 & 0 & 0 & 0 & 1/2 & (x4 - x3)/8 & 0 & 1/2 & (x3 - x4)/8 \\
1/2 & 0 & (y1 - y4)/8 & 0 & 0 & 0 & 0 & 0 & 0 & 1/2 & 0 & (y4 - y1)/8 \\
0 & 1/2 & (x4 - x1)/8 & 0 & 0 & 0 & 0 & 0 & 0 & 0 & 1/2 & (x1 - x4)/8
\end{bmatrix}
\tag{A2}
$$

$x_i$ and $y_i$ represent the coordinates of node $i$ (Figure 1) in the above equations.

## Appendix B. Weakly Coupled Bilinear Cohesive Law

*Appendix B.1. Traction–Separation Law*

This study makes use of a weakly coupled bilinear traction–separation law (TSL). In the coordinate system of the cohesive surface, it is given by

$$\mathbf{t_{coh}} = \left\{ \begin{array}{c} \sigma \\ \tau \end{array} \right\} = \begin{bmatrix} D_{11}(d_n) & 0 \\ 0 & D_{22}(d_t) \end{bmatrix} \left\{ \begin{array}{c} \delta n \\ \delta t \end{array} \right\} = [\mathbf{D_{coh/sec}}]\mathbf{\Delta u} \tag{A3}$$

where $\sigma$ (resp. $\tau$) is the normal (resp. tangential) traction and $\delta n$ (resp. $\delta t$) is the normal (resp. tangential) opening. The stiffness coefficients $D_{11}$ and $D_{22}$ depend on the normal and tangential damage variables $d_n$ and $d_t$ as follows:

$$D_{11} = \left( 1 - max\left( 0, \frac{\delta n}{|\delta n|} \right) \times d_n \right) K_n \tag{A4}$$

$$D_{22} = (1 - d_t) K_t \tag{A5}$$

$K_n$ and $K_t$ are the material parameters defining the elastic stiffness of the cohesive interface.

*Appendix B.2. Damage-Opening Relation*

Damage variable evolution is driven by the maximum normal and tangential reached openings, $\delta n^{max} = \max_{time}(\delta n)$ and $\delta t^{max} = sign(\delta t) \times \max_{time}(|\delta t|)$, which have the status of history variables. The analytical relations between the damage variables and the maximum openings are gathered in Table A1 where $\delta t^c$, $\delta t^f$, $\delta n^c$ and $\delta n^f$ are the material parameters associated with the cohesive law. The superscript $c$ stands for *critical* and indicates the opening value at which the damage starts to grow, whereas the superscript $f$ stands for *final* and designates the opening value associated with full damage. The aforesaid parameters are schematised in Figure A1.

**Table A1.** Damage-opening explicit expressions.

| | $\delta n^{max} \leq \delta n^c$ | $\delta n^c < \delta n^{max} \leq \delta n^f$ | $\delta n^{max} > \delta n^f$ |
|---|---|---|---|
| $d_n$ | 0. | $\frac{(\delta n^c - \delta n^{max})\delta n^f}{(\delta n^c - \delta n^f)\delta n^{max}}$ | 1. |
| | $|\delta t^{max}| \leq |\delta t^c|$ | $\delta t^c < |\delta t^{max}| \leq \delta t^f$ | $|\delta t^{max}| > \delta t^f$ |
| $d_t$ | 0. | $\frac{(\delta t^c - |\delta t^{max}|)\delta t^f}{(\delta t^c - \delta t^f)|\delta t^{max}|}$ | 1. |

The weakly coupled bilinear cohesive law is parametrised by the elastic stiffness parameters ($K_n$ and $K_t$) and the critical/final and normal/tangential openings (i.e., $\delta t^c$, $\delta t^f$, $\delta n^c$ and $\delta n^f$). One may also parametrise the very same cohesive law with normal and tangential critical tractions, $\sigma^c$ and $\tau^c$, as well as Mode I and Mode II critical energy release rates, $G_I^c$ and $G_{II}^c$. Indeed, explicit relations between the former and the latter sets of parameters can be written as:

$$\sigma^c = K_n \delta_n{}^c \qquad G_I^c = \int_{\delta_n=0}^{\delta_n=\delta_n{}^f} \sigma(\delta_n) d\delta_n = \frac{\delta_n{}^f \sigma^c}{2} \tag{A6}$$

$$\tau^c = K_t \delta_t{}^c \qquad G_{II}^c = \int_{\delta_t=0}^{\delta_t=\delta_t{}^f} \tau(\delta_t) d\delta_t = \frac{\delta_t{}^f \tau^c}{2} \tag{A7}$$

The weakly coupled traction–separation law (TSL) is schematically represented in Figure A1.

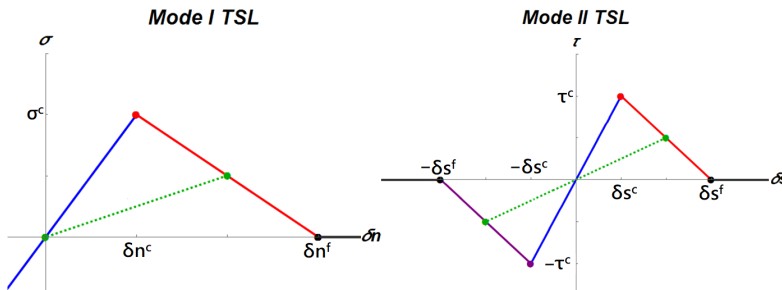

**Figure A1.** Schematic representation of the weakly coupled traction–separation law (TSL) employed in this study. The green dashed segments represent loading/unloading from a damaged state.

*Appendix B.3. Mixed-Mode Failure Criterion*

We made use of a "power law criterion" (see, e.g., [59]) to handle the mixed-mode response of the cohesive interface. The power law criterion is established in terms of an interaction equation between the Mode I and Mode II energy release rates and is given by

$$\text{If} \quad \left(\frac{G_I}{G_I^c}\right)^\alpha + \left(\frac{G_{II}}{G_{II}^c}\right)^\alpha \geq 1 \quad \text{then} \quad d_n = d_t = 1 \tag{A8}$$

where $G_I$ and $G_{II}$ are the Mode I and Mode II energy release rates of the cohesive interface and $\alpha$ is the power law exponent, which is taken to be 1 in this study.

*Appendix B.4. Tangent Cohesive Stiffness*

The incremental-iterative Newton–Rhapson scheme adopted in this study makes use of the tangent cohesive stiffness, which is given by

$$[\mathbf{D_{coh/tan}}] = \frac{\partial}{\partial \mathbf{\Delta u}}([\mathbf{D_{coh/sec}}]) = \begin{bmatrix} \frac{\partial D_{11}}{\partial \delta n} & 0 \\ 0 & \frac{\partial D_{22}}{\partial \delta t} \end{bmatrix} \tag{A9}$$

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
