# Peer review of "On the Use of Drilling Degrees of Freedom to Stabilise the Augmented Finite Element Method"

_2673-3161, doi:10.3390/applmech4040059_

Round 1

Reviewer 1 Report

Comments and Suggestions for Authors

this paper proposed a  new element which uses the Drilling Degrees of Freedom to Stabilize the Augmented Finite Element Method. The proposed element is much more effective  to simulate the dynamic crack propogation. I suggest it to be accepted for publication. However, some editorial work is still needed.

Author Response

Thank you very much for recommending our work for publication. A thorough editorial work has been performed in the revised version.

Reviewer 2 Report

Comments and Suggestions for Authors

The paper is well structured with an interesting Introduction, a clear explanation of the exotic finite element used, and a series of illustrative numerical experiences. Also, the English is good and no issues were detected. 

However, I must declare that the paper is beyond my knowledge of the finite element method. Of course, this must not undermine the work of authors, and then, based my review on the numerical part where the experiences are all interested in being convinced of its contribution to the particular task, I recommend its publishing in the journal.  

Author Response

We are glad to see that our paper is deemed interesting and well-structured. Thank you for recommending our work for publication.

Reviewer 3 Report

Comments and Suggestions for Authors

This paper presents an improved FE Method to solve non-singular stiffness matrices, which  is an  innovative improvement for the traditional  FE methods. The argumentation process of this paper is complete, the theoretical derivation is rigorous, the comparative verification is sufficient, and the results are relatively reliable. Hence, I suggest this paper could be accept with some minor modification comments.

(1)   Please explain detailly why triangular augmentation elements instead of quadrilateral augmentation elements are chosen to deal with stiffness matrix singularity.

(2)When comparing the two methods AFEM and SAFEM, the results could be organized into a table, which would be more clear to show the pros and cons of the two methods.

(3)Please unify the font format of the Figures, as shown in Figures 11,12,13,14,19,20,21 and22.

(4)In the lines567-569, the paper mentioned that "the improved accuracy of SAFEM makes up for the computational costs associated with the introduction of drilling degrees of freedom". Does this argument be proven in the article?

(5) There are to much references in the conclusion part, does it necessary?  I suggest to highlight your research finds, scope of application, innovation conclusion and further research suggestions in the conclusion part.

Comments on the Quality of English Language

No further comments.

Reviewer 4 Report

Comments and Suggestions for Authors

The submitted article can be published after its careful revision. For the details see the attached file.

Comments on the Quality of English Language

The English style is acceptable. Some minor improvements are suggested in the attached file, too.

Round 2

Reviewer 1 Report

Comments and Suggestions for Authors

This work is recommended for publication.

Reviewer 2 Report

Comments and Suggestions for Authors

I accept the paper in its present form

Reviewer 3 Report

Comments and Suggestions for Authors

No more comments.

Comments on the Quality of English Language

No comments.

Reviewer 4 Report

Comments and Suggestions for Authors

This is the second review of the same manucript.

The text of the manuscript has been corected due to the reviewer's suggestions, thus the arcticle can be recommended for publication.